# Technical note: Removing dynamic sea-level influences from groundwater-level measurements

Patrick Haehnel[1], Todd C. Rasmussen[2], and Gabriel C. Rau[3]

[1]Hydrogeology and Landscape Hydrology, Institute of Biology and Environmental Sciences, Carl von Ossietzky Universität Oldenburg, Ammerländer Heerstraße 114-118, 26129 Oldenburg, Lower Saxony, Germany
[2]Warnell School of Forestry and Natural Resources, University of Georgia, Athens GA, 30602-2152, USA
[3]School of Environmental and Life Sciences, The University of Newcastle, Callaghan, 2238, New South Wales, Australia

**Correspondence:** Patrick Haehnel (patrick.haehnel@uni-oldenburg.de)

**Abstract.** The sustainability of limited freshwater resources in coastal settings requires an understanding of the processes that affect them. This is especially relevant for freshwater lenses of oceanic islands. Yet, these processes are often obscured by dynamic oceanic water levels that change over a range of time scales. We use regression deconvolution to estimate an *Oceanic Response Function* (ORF) that accounts for how sea-level fluctuations affect measured groundwater levels, thus providing a

clearer understanding of recharge and withdrawal processes. The method is demonstrated using sea-level and groundwater-level measurements on the island of Norderney in the North Sea (Northwest Germany). We expect that the method is suitable for any coastal groundwater system where it is important to understand processes that affect freshwater lenses or other coastal freshwater resources.

## 1  Introduction

Groundwater is often the dominant source of freshwater on oceanic islands, and the sustainable management of this resource relies on understanding the gains (recharge) and losses (discharge, withdrawals) that are a function of the dynamic forces that act upon it (White and Falkland, 2009). Because freshwater on oceanic islands typically occurs as a lens above denser, saline seawater (Underwood et al., 1992), groundwater withdrawals alter fluid pressures and affect the interface between fresh- and saltwater. Excessive groundwater extraction can lead to aquifer salinization due to horizontal seawater intrusion as well as

vertical upconing (Barlow, 2003; Falkland, 1991). Thus, island groundwater resources are among the most vulnerable in the world, stressing the need for their careful monitoring and understanding to sustain their productivity (White and Falkland, 2009).

Estimating groundwater recharge on oceanic islands is challenging because groundwater levels in such systems are highly dynamic and can be influenced by multiple factors, such as periodic and aperiodic sea-level changes, coastal morphology,

aquifer properties, precipitation, and withdrawals (Jiao and Post, 2019), that interact to influence near-shore groundwater levels (e.g., Patton et al., 2021). Several methods have been used for estimating groundwater recharge, such as lysimeters (e.g., Stuyfzand, 2017), tritium-helium age dating (e.g., Houben et al., 2014; Röper et al., 2012), and stable-isotope methods (e.g.,

[18]O, [2]H, see Koeniger et al., 2016; Post et al., 2022). However, temporal differentiation of the recharge, that is critical for understanding the dynamics of coastal groundwater systems, is costly and time intensive using these methods.

Regression deconvolution provides an alternative method for quantifying groundwater processes using real-time, groundwater-level measurements. The method has been successfully applied to remove the influence of barometric pressure (Furbish, 1991; Rasmussen and Crawford, 1997), Earth tides (Toll and Rasmussen, 2007), near-surface water content (Rasmussen and Mote, 2007), and river stages (Spane and Mackley, 2011) from groundwater-level time series. Yet, despite its versatility, applications using convolution methods are commonly missing from hydrogeology textbooks (Olsthoorn, 2008). Convolution by means of

transfer function noise modeling has been applied by Bakker and Schaars (2019) to model hydraulic heads of a coastal aquifer based on time series from sea level, recharge, and groundwater withdrawal. An estimation of a response function from sea-level data itself and removal of sea-level influences from dynamic groundwater levels in coastal settings, like done with regression deconvolution, has not been performed (to the authors' knowledge). Especially in coastal settings periodic and aperiodic influences often obscure important groundwater processes, such as recharge, which is difficult to estimate or directly measure, and

pumping.

The objective of this work is to (i) provide a generic formulation for regression deconvolution, (ii) demonstrate the use of regression deconvolution for removing sea-level influences on groundwater-level measurements in an unconfined coastal aquifer consisting of unconsolidated sediments, and (iii) illustrate how the method is useful for coastal groundwater systems. The application uses groundwater-level, sea-level, and meteorologic data collected on the coastal island of Norderney, located

in Northwest Germany in the North Sea. We believe that our method is suitable for application in other coastal aquifers to support their sustainable management by better understanding the processes within – and physical characteristics of – freshwater lenses.

## 2    Influences on coastal groundwater levels

### 2.1    Conceptual overview

Figure 1 presents our conceptual model of the influence of sea levels on groundwater in coastal islands. Note that a freshwater lens is present above an underlying saltwater zone, where the depth to the freshwater-saltwater interface is a function of the water table elevation above mean sea level, as defined by the Ghyben-Herzberg principle (Jiao and Post, 2019; Post et al., 2018).

Barometric influences within unconfined aquifers are a function of the depth of the water table below the ground surface

and the air diffusivity within the unsaturated zone (Rasmussen and Crawford, 1997). Barometric pressure displays diurnal fluctuations due to solar heating, along with seasonal and weather-related forcing (McMillan et al., 2019).

Sea-level variation is dominated by diurnal and semi-diurnal periodicities, along with aperiodic behavior resulting from storm events (Boon, 2011). Further, waves breaking at the shore impact groundwater-level dynamics (e.g., Nielsen, 1999; Housego et al., 2021). Wave dynamics generally occur at high-frequencies at the shoreline (e.g., Stockdon et al., 2006; Hegge

and Masselink, 1991) while the continuous wave breaking at the shore results in a more persistent, lower-frequency wave setup

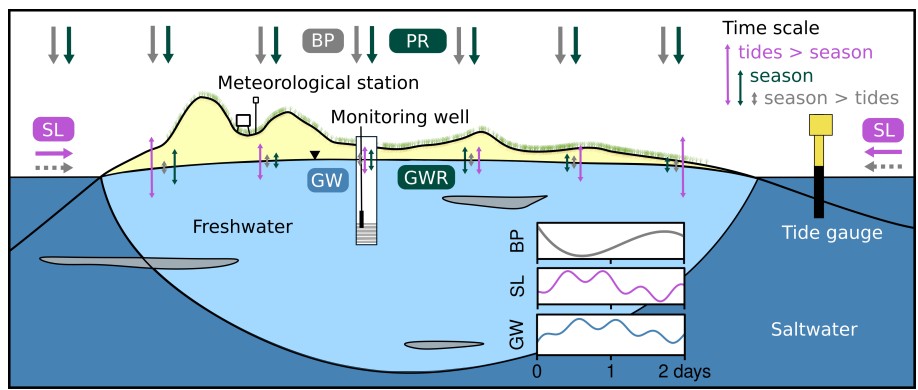

**Figure 1.** Conceptual model of groundwater-level fluctuations (GW) on a coastal island with barometric-pressure (BP), sea-level (SL), and groundwater-recharge (GWR) forcing. The latter results from precipitation (PR) on oceanic islands. Note that the amplitude of groundwater fluctuations is larger for tidal influences near the shoreline than seasonal influences, but smaller toward the center of the island. The left-hand side of the island constitutes the seaward side, while the right-hand side constitutes the leeward side of the island. Seasonal influences diminish on the leeward side of the island. Dotted-grey lines indicate the indirect influence of barometric pressure through sea levels on the groundwater levels.

(Stockdon et al., 2006; Gomes da Silva et al., 2020). Wave setup is generally larger during storm events (e.g., Senechal et al., 2011) and thus adds to the magnitude of the storm-event related, aperiodic rises in sea level.

The influence of fluctuating sea levels and waves diminishes with distance from the shoreline, with tidal and high-frequency wave variation attenuating more rapidly than variation from season, wave setup or extreme events, such as floods or droughts (Ferris, 1952; Li et al., 2004; Nielsen, 1990; Li et al., 1997; Cartwright et al., 2006; Rotzoll and El-Kadi, 2008). Precipitation recharges groundwater by vertical percolation through the overlying unsaturated zone or by direct recharge from surface-water bodies that fill during storm events.

Note, that changes in barometric pressure also affect sea level (Boon, 2011), so that barometric influence is introduced into groundwater-level time series of coastal aquifers in two principal directions: (i) vertically, through the direct influence of changes in barometric pressure, and (ii) horizontally, through the indirect influence of barometric pressure on the sea level, which is carried through the aquifer with the propagating sea-level signal (Fig. 1). Hence, the barometric influence affects groundwater levels at different time lags from the vertical and horizontal component, respectively.

## 2.2 Single-factor regression deconvolution

Barometric-pressure changes often influence groundwater levels in both confined and unconfined aquifers. The barometric efficiency (BE) is commonly used to describe the instantaneous linear relationship between discrete changes in barometric-pressure $\Delta\mathrm{BP}$ and groundwater-level responses $\Delta\mathrm{GW}$ (Rasmussen and Crawford, 1997):

$$\mathrm{BE} = -\frac{\Delta\mathrm{GW}}{\Delta\mathrm{BP}}. \tag{1}$$

While groundwater responses to barometric-pressure changes are frequently assumed to be instantaneous, there is often a delayed response that depends upon the degree of confinement, depth to the water table, borehole-storage effects, whether the borehole is open or sealed, and whether an absolute or relative (gauge) pressure sensor is used (Rojstaczer and Riley, 1990; Rasmussen and Crawford, 1997).

Response functions $\beta(\tau)$ are commonly used to quantify the time-lagged response caused by an impulse input $x(t)$ to the output time series $y(t)$ using the convolution operator $\star$

$$y(t) = \beta(\tau) \star x(t) = \sum_{k=0}^{K} \beta(\tau_k)\, x(t - \tau_k), \tag{2}$$

where $K$ is the maximum number of time lags, $t$ is the observation time, and $\tau_k = k\,\Delta t$ is the time lag between the input and the observed response, with sampling interval $\Delta t$ (Rasmussen and Mote, 2007; Rau et al., 2020). We define $m = \tau_K$, which is the maximum time lag or memory of the system beyond which the output is unaffected by an input (Rasmussen and Mote, 2007). Convolution assumes a linear, time-invariant system, with responses to individual inputs being independent of other inputs.

While convolution is used to find the output function $y(t)$ as a function of the response function $\beta(\tau)$ and the input function $x(t)$, we are often interested in finding the response function by inversion of the input and output time series using the deconvolution operator $\backslash$ (i.e., backslash)

$$\beta(\tau) = x(t) \backslash y(t). \tag{3}$$

Deconvolution can be implemented using multiple regression by forming a set of linear equations

$$y(t) = \beta(\tau_0)\, x(t - \tau_0) + \beta(\tau_1)\, x(t - \tau_1) + \cdots + \beta(\tau_K)\, x(t - \tau_K), \tag{4}$$

where the left-hand side are the observed outputs and the right-hand side consists of the unknown response function values and lagged input values (Toll and Rasmussen, 2007). This equation is written in matrix form as

$$\boldsymbol{y} = \boldsymbol{\beta}\,\mathbf{X}, \tag{5}$$

where $\boldsymbol{y}$ is the $[1 \times n]$ row vector of $n$ observed outputs, $\boldsymbol{\beta}$ is the $[1 \times m]$ row vector of unknown response coefficients, and $\mathbf{X}$ is the $[m \times n]$ matrix of observed inputs, with each row lagged by one time unit. Note that the first $m$ columns of $\boldsymbol{y}$ and $\mathbf{X}$ must be omitted unless prior input data are available; i.e., observations may be lacking for $x(t - m)$.

The resulting matrix equation can be solved using ordinary least-squares (OLS) regression, which takes the matrix form

$$\hat{\boldsymbol{\beta}} = \mathbf{X} \backslash \boldsymbol{y} = \boldsymbol{y}\,\mathbf{X}^{\mathrm{T}} \left[ \mathbf{X}\,\mathbf{X}^{\mathrm{T}} \right]^{-1}, \tag{6}$$

where the superscripts $[\cdot]^{\mathrm{T}}$ and $[\cdot]^{-1}$ indicate the matrix transpose and inverse, respectively, and where alternative matrix solvers are likely to be more efficient and accurate. The reconstructed (fitted) time series, $\hat{\boldsymbol{y}} = \hat{\boldsymbol{\beta}}\,\mathbf{X}$, can then be used to find the residual, as well as a time series that is corrected from the process influence as follows

$$\boldsymbol{y}_c = \boldsymbol{y} - \hat{\boldsymbol{y}} = \boldsymbol{y} - \hat{\boldsymbol{\beta}}\,\mathbf{X}. \tag{7}$$

The term "corrected" is used in this work and in the literature regarding regression deconvolution in the sense of "the influence of a process on the time series was removed". The use of the term "corrected" does not suggest any kind of error in the original time series.

The deconvolution was performed using first differences of the measurements, leading to Eq. (5) becoming

$$\Delta \boldsymbol{y} = \boldsymbol{\beta} \, \Delta \mathbf{X}. \tag{8}$$

This removes the effect of persistent trends in the data and therefore avoids a bias in the regression (Rasmussen and Crawford, 1997; Butler Jr. et al., 2011). To avoid spurious influences from the fact that the reconstruction hinges on an initial groundwater measurement that cannot be corrected, the mean of the corrected time series was matched to the uncorrected one.

## 2.3 Multi-factor regression deconvolution

Toll and Rasmussen (2007) and Butler Jr. et al. (2011) presented a method to analyze and remove both barometric pressure and Earth tides (i.e., two independent processes) from groundwater levels. This procedure can be extended to account for multiple drivers as follows

$$\Delta Y(t) = \sum_{p=1}^{P} \sum_{k=0}^{K^p} \beta^p(\tau_k) \, \Delta X^p(t - \tau_k). \tag{9}$$

Here, $\Delta X^p$ is the time series of the differences of influencing process $p$; $P$ represents the total number of processes; $\beta^p(\tau_k)$ represents the time-lagged impulse response function coefficients for process $p$; $m^p = \tau_{K^p}$ is the total memory for process $p$. Note that all processes propagate through the subsurface either vertically or horizontally, and are increasingly attenuated and time-lagged with distance from their origin. This approach allows us to consider multiple dynamic processes that could affect groundwater levels, including precipitation, evapotranspiration, barometric pressure, streamflow, Earth tides, soil moisture, etc. Note that process-based indices are always notated as superscripts here.

## 2.4 Process response functions and time series correction

The response function for a process is determined from the impulse responses (Eq. 6) as follows

$$B^p(\tau_k) = \sum_{k=0}^{K^p} \hat{\beta}^p(\tau_k). \tag{10}$$

Note that we state the process response function $B^p$ as a generic term that allows disentanglement of multiple processes $p$ each with total memory $m^p$. For example, the *Barometric Response Function* (BRF) is determined by taking the cumulative sum of the impulse responses to barometric pressure, $\hat{\beta}^{BP}$ (Rasmussen and Crawford, 1997)

$$\mathrm{BRF}(\tau_k) = \sum_{k=0}^{K^{\mathrm{BP}}} \hat{\beta}^{\mathrm{BP}}(\tau_k). \tag{11}$$

Analogously, an *Earth Tide Response Function* (ETRF) as well as a *River Response Function* (RRF) can be formulated in the same way. These influences have successfully been used to characterise subsurface processes and properties and to correct

groundwater levels from the respective influences (e.g., Spane, 2002; Toll and Rasmussen, 2007; Butler Jr. et al., 2011; Spane and Mackley, 2011; Rau et al., 2020). Here, we note that despite being used to correct groundwater levels, the name ETRF has not explicitly been defined in the literature.

The aim of this work is to illustrate how regression deconvolution can be used to estimate the *Oceanic Response Function* (ORF)

$$\text{ORF}(\tau_k) = \sum_{k=0}^{K^{\text{SL}}} \hat{\beta}^{\text{SL}}(\tau_k). \tag{12}$$

This characterizes the effects of sea-level fluctuations $\text{SL}(t)$ on measured groundwater levels

$$\text{GW}(t) = \text{ORF}(m^{\text{SL}}) \star \text{SL}(t), \tag{13}$$

with sea-level memory $m^{\text{SL}}$. We note that our approach employs multi-factor regression deconvolution to disentangle the simultaneous influences of sea levels and barometric pressure on observed groundwater levels, so processes $p = \{\text{SL}, \text{BP}\}$ in Eq. (9). We did not analyze Earth-tide responses, as they are generally negligible in unconfined finite-depth aquifers made of unconsolidated sediment (Rojstaczer and Riley, 1990). The formulated correction procedure yields corrected groundwater levels

$$\text{GW}_c(t) = \text{GW}(t) - \sum_{p=1}^{P} \sum_{k=0}^{K^p} \hat{\beta}^p(\tau_k) \, \Delta X^p(t - \tau_k). \tag{14}$$

Again, the mean of the corrected values must be matched to the mean of the uncorrected values (as explained earlier).

A wave response function and groundwater levels with wave setup removed can be obtained equivalently, e.g. to account for additional storm-event related wave setup at the shore. Alternatively, wave setup can be incorporated into the sea-level time series to obtain an ORF representing both processes. Note that wave setup is generally estimated from offshore wave measures (e.g., Gomes da Silva et al., 2020).

Besides regression deconvolution, transfer function noise models are used to model groundwater-level time series from time series of stresses (e.g., groundwater recharge, groundwater extraction, sea levels) using convolution (e.g., von Asmuth et al., 2002; Collenteur et al., 2019; Bakker and Schaars, 2019) and to estimate unknown stresses from groundwater-level time series (e.g., Collenteur et al., 2021; Pezij et al., 2020). The method differs from regression deconvolution in that the response function is pre-defined with a fixed shape, typically by a probability density function like the Gamma distribution (Collenteur et al., 2019), and not obtained through the data itself.

## 2.5 Considering density effects

The density difference between seawater and freshwater has to be considered when applying Eqs. (8), (9), and (14) with sea levels present in $\Delta \mathbf{X}$. Here, the ORF is defined based on hydraulic head measurements in freshwater. The propagation of external influences in the aquifer depends on the pressure of the external stressor rather than the elevations, which are used as a proxy (i.e. hydraulic heads). A change of hydraulic head in seawater yields a larger pressure change than the same hydraulic

head change in freshwater would due to the density difference. Therefore, sea level records need to be corrected for this higher density to correctly represent the pressure changes of the sea level at the shore with reference to fresh groundwater inland.

Density correction of hydraulic heads is typically achieved by calculating freshwater heads

$$h_f(t) = \frac{\rho}{\rho_f} h(t) - \frac{\rho - \rho_f}{\rho_f} z, \tag{15}$$

where $h$ is the measured point water head, $\rho_f$ is the freshwater density ($1000 \ \mathrm{kg \, m^{-3}}$) and $\rho$ is the density of the water at the screen elevation $z$ of a monitoring well (Post et al., 2007). In case of sea-level observations, $\rho$ is the seawater density and $z$ is the elevation of the sea floor. When using first differences, the freshwater head difference between times $t_i$ and $t_{i-1}$ is

$$\Delta h_f = h_f(t_i) - h_f(t_{i-1}) = \frac{\rho}{\rho_f} [h(t_i) - h(t_{i-1})] = \frac{\rho}{\rho_f} \Delta h \tag{16}$$

so that sea-level differences in Eqs. (9) and (14) have to be defined as freshwater-equivalent differences

$$\Delta X_f^{\mathrm{SL}} = \frac{\rho}{\rho_f} \Delta X^{\mathrm{SL}} \tag{17}$$

which corrects differences from measured sea levels $\Delta X^{\mathrm{SL}}$ by the density ratio $\rho/\rho_f$ between salt- and freshwater.

Should the groundwater monitoring well be screened in a location of brackish water or saltwater, the density correction needs to be applied to the hydraulic head differences as well to obtain freshwater-equivalent hydraulic head differences

$$\Delta Y_f(t) = \frac{\rho(t)}{\rho_f} \Delta Y \tag{18}$$

which allows to obtain comparable ORFs between monitoring sites. Especially at beach sites, the density ratio may be a function of time reflective of salinity changes over time around the screen of the monitoring well (Grünenbaum et al., 2023; Greskowiak and Massmann, 2021). Details on the estimation of groundwater density from electric conductivity measurements are provided by Post (2012).

## 3 Application

### 3.1 Field site, monitoring, and data processing

Norderney is a coastal barrier island that is part of the East Frisian island chain located in the North Sea near the Northwest German coast (Fig. 2). The island covers an area of about $25 \ \mathrm{km^2}$, with an east-to-west extent of 14 km and an average north-to-south extent of 2 km (Naumann, 2005; Streif, 1990). Rainfall is the only source of freshwater on the island, and 782 mm of precipitation were observed during our one-year research period (1 November 2018 to 31 October 2019) at the

185 Norderney meteorological station (DWD Climate Data Center (CDC), 2021a). Approximately half of the island's precipitation was estimated to recharge the aquifer (Naumann, 2005).

Semi-diurnal tides dominate Norderney's sea-level fluctuations. For our research period, the mean high water (MHW) was 1.26 m asl (above sea level), and the mean low water (MLW) was -1.18 m asl (Wasserstraßen- und Schifffahrtsamt Ems-Nordsee (WSA Ems-Nordsee) [Waterways and Shipping Authority Ems-North Sea], 2021), which yields a tidal range of 2.44

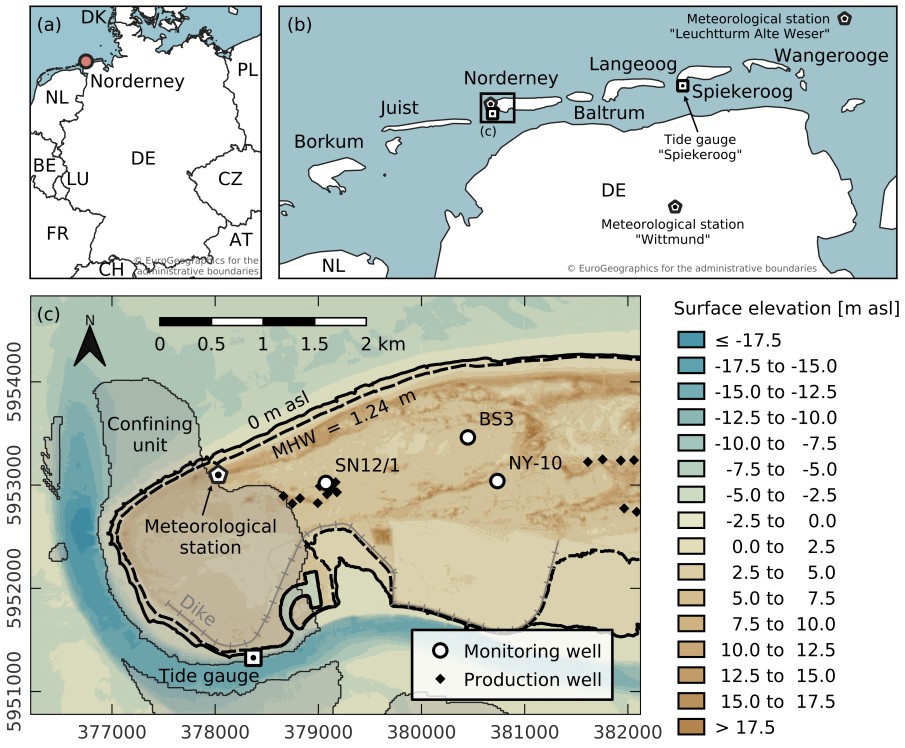

**Figure 2.** Map of Norderney Island in Northwest Germany showing three monitoring wells, production wells, tide gauge, meteorological station, and a confining unit (shaded area) to the west. Mean high water (MHW) is the average between 2010 and 2020 (WSA Ems-Nordsee, 2021). Coordinate reference system is UTM Zone 32N (EPSG:25832). Data sources: EuroGeographics and UN-FAO (2020), © EuroGeographics for the administrative boundaries; Haehnel et al. (2023); Niedersächsischer Landesbetrieb für Wasserwirtschaft, Küsten- und Naturschutz (NLWKN) [Lower Saxony Water Management, Coastal Defense and Nature Conservation Agency] (2021); Sievers et al. (2020); Stadtwerke Norderney (2021b); WSA Ems-Nordsee (2021).

m that corresponds to meso-tidal conditions (Hayes, 1979). Seasonal flooding typically occurs during the autumn and winter seasons (Holt et al., 2019) and is defined using a sea level 1.5 m above MHW for the region (Gönnert, 2003). The maximum sea level during our study period was 3.03 m asl (1.77 m above MHW) on 8 January 2019 (WSA Ems-Nordsee, 2021).

The island's geomorphology is characterized by beaches and dunes on the seaward north, and salt marshes and back-barrier tidal flats on the leeward south (Petersen et al., 2003). Holocene dune sediments are composed of fine-grained sands and sand flat with mixed flat deposits, extending to about 30 to 40 m bsl (below sea level) in the central part of the island (Naumann, 2005; Streif, 1990). These sediments extend to a depth of about 10 m bsl below the western part of the island, where they transition to a confining unit of Holocene clay, silt, and basal peat (Schaumann et al., 2021), shown in Fig. 2c. Mud flat deposits are present locally below the central part of the island (Naumann, 2005). Pleistocene sandy deposits are found below Holocene sediments, which largely originated from Drenthian sandur-type plains (Naumann, 2005; Schaumann et al., 2021).

**Table 1.** Reference data for the groundwater monitoring wells (Stadtwerke Norderney, 2021b). Coordinate reference system is UTM Zone 32N (EPSG:25832).

| | Well name | | |
|---|---|---|---|
| | BS3 | NY-10 | SN12/1 |
| Latitude [°N] | 53.716 | 53.712 | 53.712 |
| Longitude [°E] | 7.188 | 7.193 | 7.168 |
| Northing [m] | 5 953 462 | 5 953 039 | 5 953 021 |
| Easting [m] | 380 449 | 380 736 | 379 073 |
| Ground surface elevation [m asl] | 2.50 | 2.83 | 4.48 |
| Average groundwater table [m asl][a] | 1.57 | 1.86 | 1.23 |
| Average depth to water table [m][a] | 0.93 | 0.97 | 3.25 |
| Top of screen [m asl] | -4.98 | -3.57 | -18.02 |
| Bottom of screen [m asl] | -6.98 | -4.57 | -20.02 |
| Screen length [m] | 2 | 1 | 2 |
| Casing diameter [cm] | 5 | 5 | 5 |
| Distance to 0 m asl [m][b] | 741 | 1154 | 688 |
| Distance to MHW [m][c] | 692 | 979 | 456 |
| Distance to production well [m][d] | 1187 | 896 | 39 |

[a] Averaged over the studied time frame from 1 November 2018 to 31 October 2019.
[b] Minimum Euclidean distance to 0 m asl contour using a DEM of Sievers et al. (2020).
[c] Minimum Euclidean distance to mean high water (MHW) contour (1.24 m asl, average between 2010 and 2020 from WSA Ems-Nordsee (2021)) using a DEM of Sievers et al. (2020).
[d] Euclidean distance to closest production well.

A more detailed summary of the island's development, geomorphology, geology, and hydrogeology can be found in Haehnel et al. (2023). Schaumann et al. (2021) described the Holocene and Pleistocene geology in detail, and Karle et al. (2021) reconstructed the Holocene landscape development of the area during sea-level transgression.

Hourly groundwater levels are routinely collected by the Municipal Works Norderney using STS DL/N 70 dataloggers in open (uncapped) monitoring wells (Stadtwerke Norderney [Municipal Works Norderney], 2021a). This study focuses on a subset of these wells (SN12/1, BS3, NY-10) for the one-year period between 1 November 2018 and 31 October 2019. At the given time series length of one year, time increments of one hour are generally sufficient to capture the tidal constituents present at the study site (Schweizer et al., 2021). As summarized in Table 1, the monitoring wells have short (1 to 2 m) screen lengths. Both BS3 and NY-10 screened zones are shallow, while SN12/1 has a deeper screen from 18 to 20 m bsl, which is below the base elevation of the nearby confining unit (Fig. 2c; Haehnel et al., 2023). All three observation wells are screened entirely in the freshwater lens of the island. Both SN12/1 and BS3 are located at similar straight-line distances (688 and 741 m, respectively) to the shoreline (i.e., the 0 m asl contour line), while NY-10 is located more centrally on the island at a greater distance (1154 m) (Table 1). The distance to the MHW contour line is also presented in Table 1 because the shoreline distance is ambiguous when tides are present.

Hourly barometric-pressure and precipitation data were obtained from the meteorological station located near the north-western shoreline (DWD Climate Data Center (CDC), 2021b, c). The spatial distance between the meteorological station and the groundwater monitoring wells is approximately 1 km in case of SN12/1 and approximately 2.5 km in case of BS3 and NY-10. At this distance, the barometric pressure observations are assumed to be representative for the groundwater monitoring locations as the barometric pressure typically varies at larger spatial scales (cf. Appendix A). Daily precipitation totals are used for graphical comparison with other variables (DWD Climate Data Center (CDC), 2021a).

Sea levels collected at one-minute intervals were obtained from the tide gauge "Norderney Riffgat" (Wasserstraßen- und Schifffahrtsverwaltung des Bundes (WSV) [Federal Waterways and Shipping Administration], 2021a), located near the south-western shoreline. Tidal data were downsampled to hourly intervals for subsequent analysis by discarding observation time points that did not match the sampling times of groundwater and barometric pressure data, which were collected at each full hour. Sea-level differences as required for Eq. (14) were converted to freshwater-equivalent sea-level differences according to Eq. (17) with density ratio $\rho/\rho_f = 1.025$, assuming a saltwater density of $1025 \ \mathrm{kg\,m^{-3}}$ at the study site. The spatial distance of the tide gauge from the shoreline segments closest to the observation wells should not affect the results presented here, because the temporal offset of the sea-level signal at these shoreline segments compared to the tide gauge is in the order of a few minutes, much shorter than the sampling interval of 1 h used in this study (cf. Appendix A). An hourly time series of the extracted water volume from the western production well cluster near SN12/1 (Fig. 2c) between 13 and 20 November 2022 was provided by the local water supplier (Stadtwerke Norderney, 2023).

Groundwater and tidal data were inspected prior to analysis and no issues (e.g., gaps, spikes, steps) were found. Barometric-pressure and precipitation data were examined using an automated evaluation and correction procedure by the data provider (DWD Climate Data Center (CDC), 2021a, c, b). No data are missing in any time series related to Norderney during the research period. All data were converted to time zone UTC+1.

The low-pass finite-impulse-response filter "LP241H079122kM3" from Shirahata et al. (2016) was applied to groundwater and sea levels for comparison with regression deconvolution results. The filter uses a ten-day symmetric window designed to remove diurnal and semi-diurnal tidal constituents as well as their higher harmonics.

## 3.2    Processes affecting groundwater levels

Sea-level, barometric-pressure, and daily-precipitation data are presented in Fig. 3a and 3b. Note the aperiodic meteorological
as well as the sea-level influences, that are dominated by astronomical tides, on groundwater levels (Fig 3c–e). This demonstrates the overlapping effects of both vertical propagation of atmospheric effects as well as lateral effects of sea-level variation. Groundwater levels show an oscillating semi-diurnal pattern with differing magnitudes due to sea-level influences that propagate through the aquifer (Fig. 3c–e) and reflect both periodic as well as aperiodic changes in sea level (e.g., the storm event on 8 January 2019). The well furthest from the shoreline, NY-10, shows the strongest attenuation of the oscillating sea levels,
while the attenuation in BS3 and SN12/1 is smaller due to their greater proximity to the shoreline. Yet, BS3 is more strongly attenuated than SN12/1 despite their similar distance to the shoreline. This is likely explained by the nearby confining unit in the west (Fig. 2) that allows the signal to propagate more rapidly due to a smaller storativity.

In addition to changes in sea level, groundwater levels in BS3 and NY-10 show precipitation responses, but these are largely obscured in SN12/1. The precipitation response of BS3 and NY-10 is discernible in mid-August 2019, where groundwater
levels increase despite a lack of change in sea levels. Also note that groundwater levels increase while sea levels decrease in late-September 2019 to early-October 2019.

## 3.3    Removing dynamic sea-level influences

Periodic and aperiodic sea-level as well as barometric-pressure fluctuations were removed from groundwater-level measurements using regression deconvolution (Fig. 3c–e). The storm event on 8 January 2019 provides an opportunity to evaluate
our method. Here, the original groundwater-level time series and their trend (Fig. 3c–e) react to the sudden increase in sea level. The corrected time series now shows only a minor response to the storm event with small increase that is likely due to storm-related recharge and wave setup.

Corrected groundwater levels in BS3 and NY-10 now show contemporaneous responses to precipitation events that increase with increasing precipitation (Fig. 3cd). For example, the precipitation response is now readily observed in early-April 2019,
mid-August 2019, and late-September 2019. Note further that corrected groundwater levels remove more of the sea-level influence than filtered trends (e.g., March 2019). The corrected signal now provides a useful tool for examining the duration and magnitude of groundwater recharge.

While regression deconvolution assumes a linear response of groundwater levels to the external influence (cf. Sect. 2.2), i.e., sea levels, it can be assumed that the response to ocean tides is nonlinear due to the changes in aquifer thickness and the
low-pass filter effect of the aquifer sediment (e.g., Nielsen, 1990; Rotzoll et al., 2008). The latter causes amplitudes of lower-frequency tidal constituents to be attenuated less with increasing distance to the shoreline than higher-frequency ones (Trefry and Bekele, 2004). Further, the phase shift of lower-frequency tidal constituents in the sediment is slower than for higher-

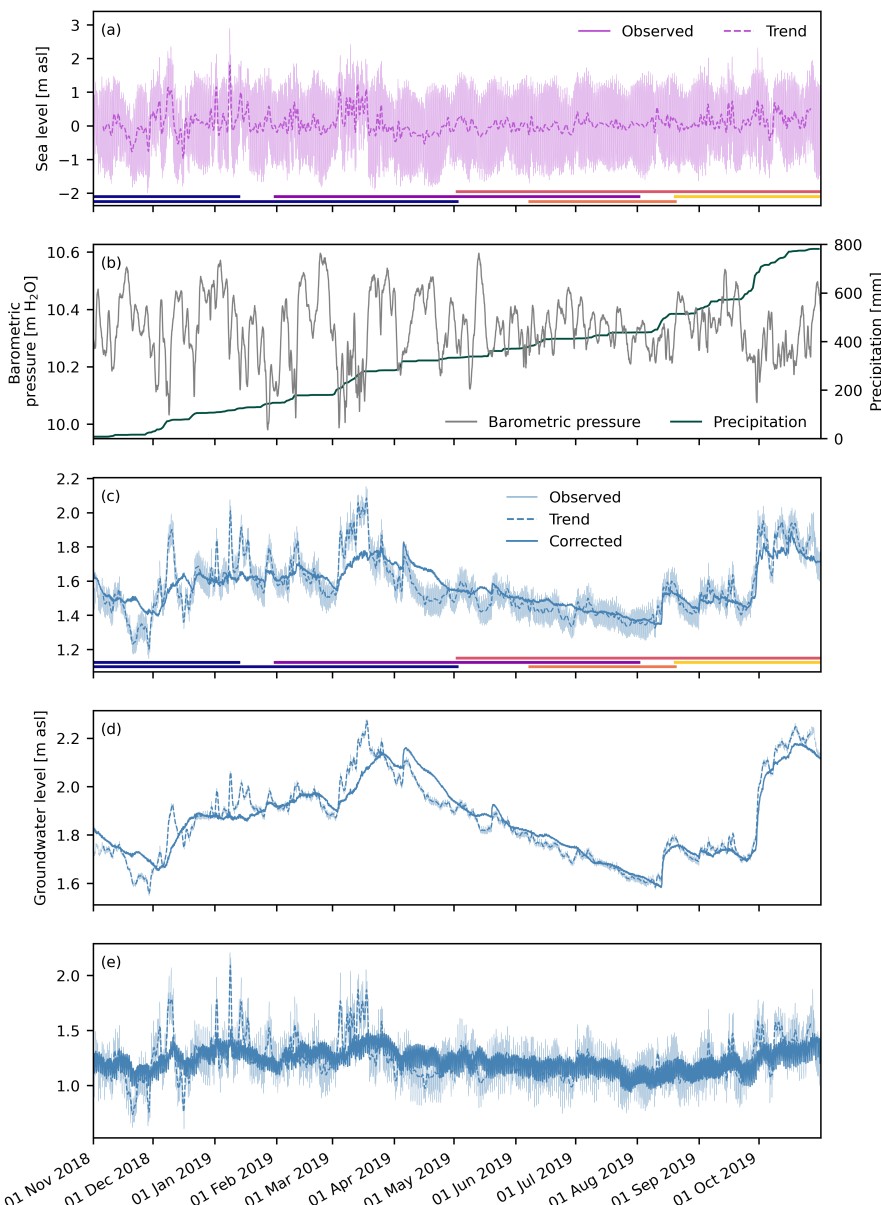

**Figure 3.** Time series of (a) sea level, (b) barometric pressure and cumulative daily precipitation, and observed as well as corrected groundwater levels in (c) BS3, (d) NY-10, and (e) SN12/1. Oceanic Response Function memories $m^{\mathrm{SL}}$ are 150, 250, and 48 h, respectively, while the Barometric Response Function memory $m^{\mathrm{BP}}$ is 24 h for each monitoring well. Trends of sea and groundwater levels are shown as well in (a) and (c-e). Colored, horizontal bars in (a) and (c) indicate the time frames covered by shorter portions of the time series for which the ORF was calculated (Fig. 5).

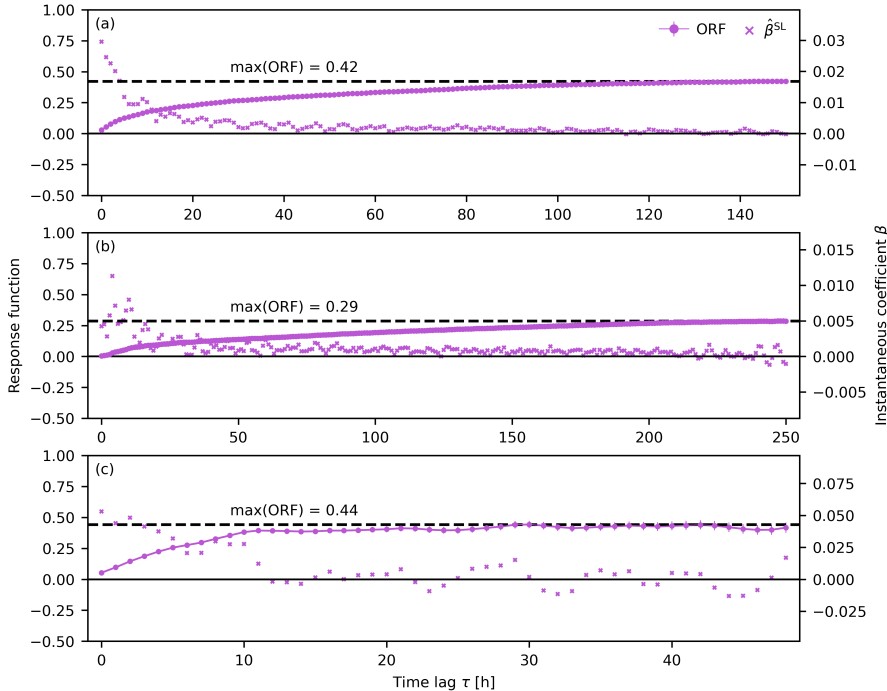

**Figure 4.** Oceanic Response Function (ORF) for (a) BS3, (b) NY-10, and (c) SN12/1 with with corresponding instantaneous coefficients $\hat{\beta}^{\mathrm{SL}}$. Note the different maximum time lag for each well on the x-axis. Vertical error bars indicate an uncertainty of one standard error for the Oceanic Response Function (Appendix B).

frequency ones (Rotzoll et al., 2008). Additionally, higher-harmonic tidal constituents (i.e., shallow water tidal constituents) are generated within the aquifer sediment introducing another source of nonlinearity (Bye and Narayan, 2009).

Smith (2008) reported that linear approximations for periodic flow can be adequate if the changes in saturated aquifer thickness were comparably small, and Reilly et al. (1987) stated a temporal variability of maximum 10 % as a rule of thumb for nonlinear influences. With a tidal range of 2.44 m asl (cf. Sect. 3.1) and an approximate aquifer thickness of 400 to 450 m (Haehnel et al., 2023), linear approximation seems valid here.

Wave setup was not considered as a separate process since the additional considerations required for an empirical formula
to estimate wave setup from offshore measures (Gomes da Silva et al., 2020) were beyond the methodological objective of this technical note. The influence of wave setup on groundwater levels may however be present in the corrected time series when the wave setup present at calm conditions increases during storm events for example (i.e. wave setup is not constant over the studied time frame; cf. Section 2.1). Here, this could be the case during the storm event in January 2019 or the time frame of pronounced sea-level variations in March 2019 for example (Fig. 3).

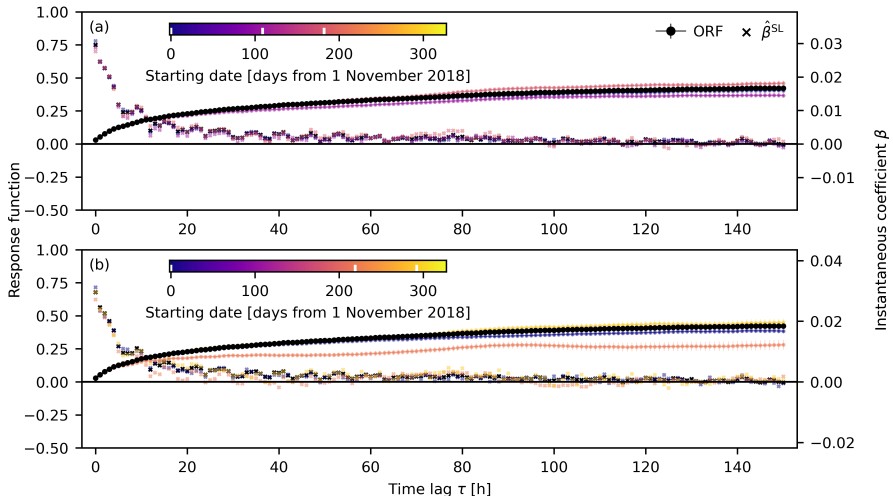

**Figure 5.** Oceanic Response Function (ORF) for BS3 with a time series length of (a) 182.5 days and (b) 73 days. The black response functions and instantaneous coefficients show the results of the analysis of the complete one-year time series (Fig. 4a). Colors indicate the time difference of the starting point of the shorter time series from the starting point of the complete time series. Note that not all analyzed response functions are shown (cf. to Figs. S5 and S8 in the supplement). The starting dates of the analyzed time series are displayed by white stripes in the colorbars. Time frames covered by the time series corresponding to the ORFs shown are displayed in matching colors in Fig. 3ac.

### 3.4 Response Functions

Figure 4 shows the instantaneous coefficients $\hat{\beta}^{\mathrm{SL}}$ and their cumulative sum that represents the Oceanic Response Function (ORF). The coefficients are largest for small time lags and approach zero at longer lag times. Note that values should approach zero as they approach the memory of the system (i.e., sea-level changes no longer influence groundwater levels). Also note that each well has a unique ORF which can also vary with time as a result of temporally variable characteristics of the sea-level influence (Brookfield et al., 2017). Similar to the river-stage response function used by Spane and Mackley (2011), this memory should be longer for locations further from the source. The ORF is greater for stronger influences than for weaker influences, which is also a function of the distance to shoreline. Similar to the river-stage response function, the ORF is a function of aquifer hydraulic diffusivity, shoreline distance, beach sediment composition, borehole-storage, and well-skin effects (Spane and Mackley, 2011).

The maximum lag time (i.e., memory) also varies by well, with 150 and 250 h for BS3 and NY-10, respectively, which reflects the greater distance to the shoreline of NY-10. The ORF stabilizes to maximum values of 0.42 and 0.29 for BS3 and NY-10, respectively (Fig. 4ab), again reflecting the distance to the shoreline. The Harmonic Least Squares (HALS) analysis applied to corrected time series with different sea-level memories suggests that ocean tides are removed with small lags, and longer lags are required for aperiodic events (Appendix C).

Besides the distance to coast and aquifer hydraulic properties, the characteristics of the sea-level fluctuations within the analyzed time period are relevant for the shape of the ORF (Brookfield et al., 2017). For Norderney, the most prominent change in sea-level characteristics is the presence of storm floods during the winter half year and the general lack of them during the summer as well as the generally higher variability of non-tidal sea-level components during winter and autumn (trend line in Fig. 3a). Figure 5 shows ORFs calculated for subsets of the one-year time series of 182.5 days and 73 days with different starting dates (time frames covered are indicated in Fig. 3ac). The ORFs are relatively close in shape to the ORF of the entire time series, when winter and/or autumn are covered, i.e., they cover either the start or end of the one-year period. When no time frame with pronounced variability in the non-tidal sea-level component is covered, the maximum ORF value tends to be smaller than that of the entire time series (orange line in Fig. 5b, Figs. S1 to S9 in the supplement), which resembles the then weaker influence of sea-level fluctuations on the groundwater levels (see Appendix D for more details). This is not observed for the 182.5-day time series, as all of them cover a time frame with pronounced non-tidal sea-level variability.

In the case of the 73-day time series starting in early-June 2019 (orange line in Fig. 5b), sea levels show no pronounced variation besides ocean tides (Fig. 3a). Accordingly, the instantaneous coefficients start fluctuating around zero earlier than for the other time series (Fig. 5b), indicating a shorter memory $m^{\mathrm{SL}}$ of around 48 h. In conclusion, the ORF seems to be time invariant as long as the characteristics of the stresses covered by the individual time series are comparable.

Note, that generally the maximum number of time lags (i.e., number of instantaneous coefficients) used in the regression deconvolution should only constitute a small portion of the number of time steps present in the analyzed time series to avoid overfitting. Thus, systems with longer memory require longer time series to produce meaningful response functions. In our case for example, the longest required memory of 250 h is around 3 % of the one-year time series.

The *Barometric Response Function*s (BRF) for BS3 and NY-10 deliver small values and instantaneous coefficients start fluctuating around zero for $\tau > 0$ h for BS3 and $\tau > 4$ h for NY-10 (Fig. 6ab). Thus, the response to barometric-pressure changes is instantaneous (smaller than the measurement interval, BS3) or relatively fast (NY-10). This is consistent with shallow water tables and high air permeabilities in the sandy surficial deposits that promotes rapid equilibration of aquifer heads (cf. average depth to water table in Table 1) (Rasmussen and Crawford, 1997).

Well SN12/1 shows a faster response to sea-level changes and the maximum ORF of 0.44 is attained within two days (Fig. 4c), which can be explained by the presence of the nearby confining unit (Fig. 2c). However, corrected groundwater levels still show periodic fluctuations (Fig. 3e) that HALS analysis identified as a diurnal pattern associated with the $S_1$ tidal constituent that is not removed by deconvolution because it is not present in sea-level observations (Fig. C2c). This $S_1$ response may be due to meteorological (e.g., evapotranspiration) or other (e.g., groundwater extraction) influences that vary at this frequency (cf. Sect. 3.5). Further, the BRF of SN12/1 shows a pronounced periodic pattern at ca. 2 cpd.

## 3.5 Revealing groundwater extraction and aquifer-generated tidal constituents

Figure 7 shows an eight-day window in November 2018 of observed and sea-level corrected groundwater levels from SN12/1. While the influence of groundwater extraction was masked by sea-level influences, it is clearly present after correction. Groundwater declines in the corrected time series coincide with daily extraction. This explains the visible mixed-tide type present in

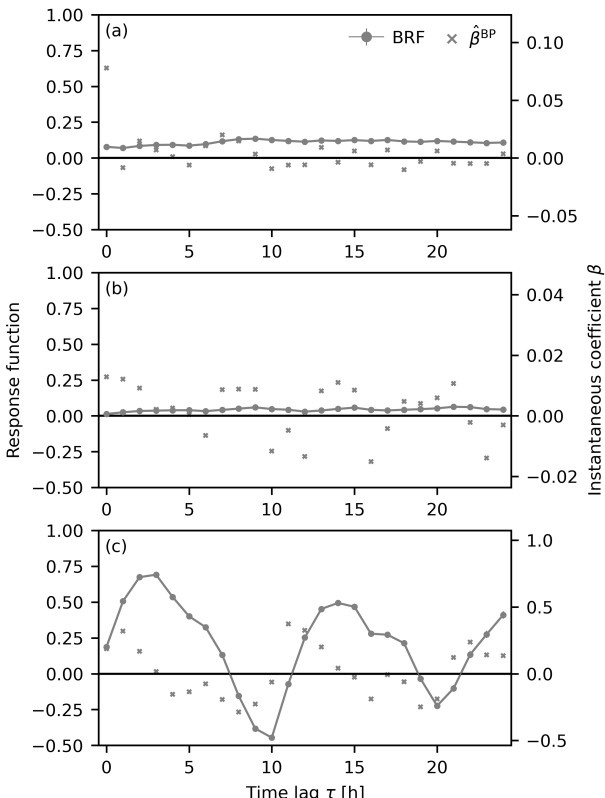

**Figure 6.** Barometric Response Functions (BRF) for (a) BS3, (b) NY-10, and (c) SN12/1 with with corresponding instantaneous coefficients $\beta^{\mathrm{BP}}$. Vertical error bars indicate an uncertainty of one standard error for the Barometric Response Function (Appendix B).

observed groundwater levels that cannot originate from the semi-diurnal, $M_2$-dominated ocean tide, with only small diurnal
components (cf. Fig. C1).

    We compare this pattern with groundwater extraction data from 2022, which shows that pumping patterns are similar to corrected groundwater levels. We rely on 2022 extraction data because such data were not available during the study period. Also, seasonal extraction patterns and yearly extraction volumes have remained stable since the early 2000s (Stadtwerke Norderney, 2021b). The strong coherence between these two time series provides further evidence for the utility of regression
deconvolution for removing interference from external stimuli.

    While the pattern of groundwater extraction is clearly visible in the groundwater-level time series of SN12/1, this influence is also present at monitoring wells BS3 and NY-10. To show this, amplitudes of frequencies between 0 and 12 cpd were extracted from the corrected groundwater-level time series using HALS analysis (cf. Appendix C) and the Fast Fourier Transform (FFT) with a Hanning window (Fig. 8). This shows the daily groundwater extraction pattern strongly enhances the $S_1$ tidal constituent
at SN12/1 to around 6 cm (Fig. 8c) compared to an amplitude of around 0.8 cm present in the ocean-tide signal (Fig. C1). For

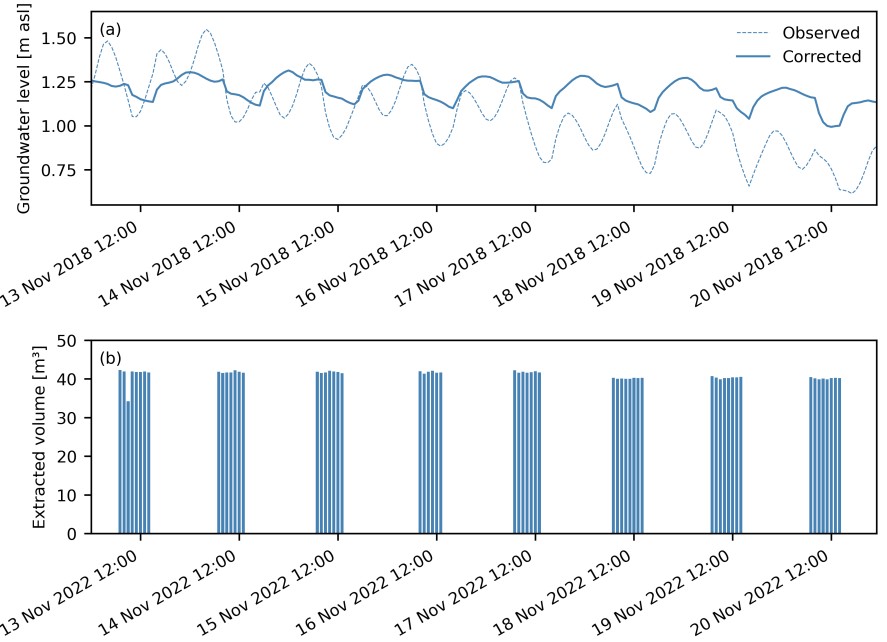

**Figure 7.** Time series of (a) observed and corrected groundwater levels of SN12/1 and (b) extracted groundwater volume of the production wells around SN12/1 for an eight-day time period. Note that the years of both time series differ since no hourly extraction data were available for the studied time frame. However, overall groundwater extraction patterns over a season are generally stable and comparable since the early 2000s (Stadtwerke Norderney, 2021b) so that a main extraction time period between 7 AM and 3 PM is very likely for 2018 as well.

BS3 and NY-10, the amplitude of tidal constituent $S_1$ introduced by groundwater extraction is much smaller due to the larger distance to the production wells (Fig. 8ab).

The groundwater extraction signal is one of the causes of the small amplitude, high-frequency oscillations remaining in the corrected groundwater level time series (Fig. 3ab) and the oscillation visible in the instantaneous coefficients of the regression
deconvolution (at ca. 4 cpd, cf. Fig. 4ab) of BS3 and NY-10. The second source of the oscillations is the generation of the shallow water tidal constituent $M_4$ within the aquifer as a result of the propagation of the tidal signal in the sediment (Bye and Narayan, 2009, cf. Sect. 3.3). It is generated as the higher harmonic of the $M_2$ constituent, which is the dominating tidal constituent at the study site (Fig. C1), so that the amplitude and phase lag of the generated $M_4$ constituent depend on the amplitude and phase lag of the ocean tide $M_2$ constituent (Bye and Narayan, 2009). For the large amplitude of the ocean
tide $M_2$ constituent (Fig. C1), the amplitude of the generated $M_4$ constituent is still discernible from noise in the data at the monitoring wells (Fig. 8). Further, there is noise present in the corrected time series for frequencies between 0.5 and 3 cpd which cannot be attributed to major tidal constituents (Fig. 8), but parts of it may be attributed to the frequency-dependent amplitude attenuation and phase shift of the different tidal constituents within the aquifer sediment (cf. Sect. 3.3).

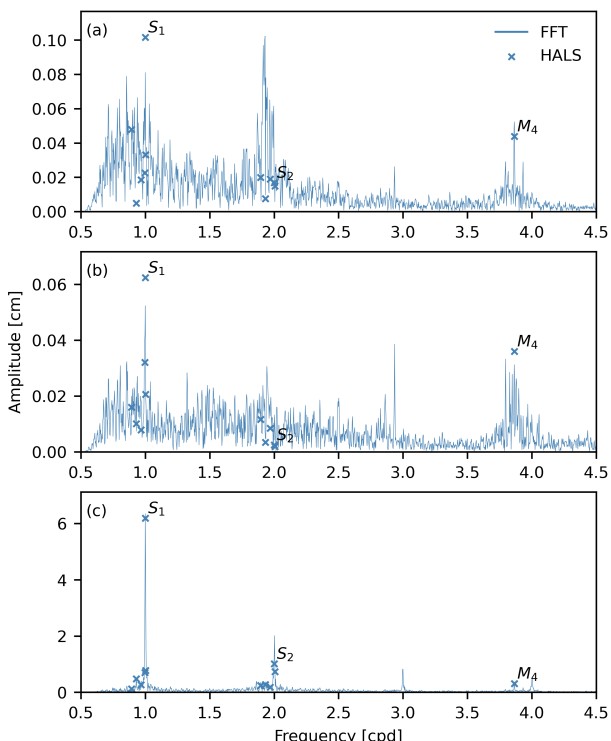

**Figure 8.** Amplitudes found in the corrected groundwater-level time series for frequencies between 0.5 and 4.5 cpd obtained with Harmonic Least Squares (HALS) analysis and Fast Fourier Transform (FFT) for (a) BS3, (b) NY-10, and (c) SN12/1. The HALS data shows tidal constituents as outlined in Fig. C1. Note the different y-axis scales on each panel.

The oscillating BRF of SN12/1 (Fig. 6c) is likely a result of the groundwater extraction signal not being present in the regression deconvolution. A similar pattern was observed by Patton et al. (2021) in their analysis of barometric-pressure and Earth-Tide response of groundwater levels in a coastal aquifer regarding ocean tides (they termed this shape "peaked"). In their study, they did not consider sea-level fluctuations and the semi-diurnal ocean tide pattern mapped to the BRF. The oscillation in the BRF of SN12/1 is likely mixed semi-diurnal/diurnal because the tidal constituents $S_1$ and $S_2$ introduced by groundwater extraction are not removed at the given memory $m^{\mathrm{SL}} = 48$ h (Fig. C2c).

## 4  Conclusions

We demonstrate how regression deconvolution can be used to remove sea-level influences from groundwater levels measured in coastal aquifers, which has not been illustrated before. We define and use an Oceanic Response Function (ORF) to represent the time lag dependent response coefficients for characterizing groundwater responses to sea-level changes. Once sea-level

influences have been removed, the resulting groundwater levels clearly show previously masked responses to precipitation and groundwater extraction. In this application, the horizontal propagation of sea-level changes dominates groundwater responses.

Our findings expand the range of applications for regression deconvolution by enabling the characterization and mitigation of external perturbations impacting groundwater levels. These perturbations encompass barometric pressure, Earth tide, river stage fluctuations, and now, oceanic influences. Our methodology is well-suited for analyzing data obtained from groundwater monitoring in oceanic and coastal aquifers. This capability is instrumental in enhancing our understanding and sustainable management of these critical water systems. Future research endeavors should prioritize a systematic exploration of how hydraulic processes (e.g., modulation of tidal signals within aquifer sediments) and properties (e.g., hydraulic diffusivity) in coastal aquifers affect Oceanic Response Functions. Additionally, estimating response functions linked to groundwater extraction becomes an important area for investigation once suitable data becomes available.

The ORF shape depends on the stresses being present in the sea-level time series and are only similar for different time frames, i.e., time invariant, when the stresses of the time frames are similar. A time frame containing storm events may yield a different ORF than a time frame were ocean tides are the most prominent sea-level influence. In the case of Norderney, the assumption of a linear response of groundwater levels to sea-level influences will likely be valid approximately resulting from the small changes in saturated aquifer thickness introduced by the sea-level fluctuations (Reilly et al., 1987; Smith, 2008).

While many hydrogeological settings will likely require the estimation of other effects (e.g., Earth tides, soil moisture, river stage), we neglect these influences at this site due to their minimal influence. Regardless of the specific application, however, our methodology for removing multiple factors should provide sufficient flexibility for interpreting and removing these influences.

## Appendix A: Spatial variability of barometric pressure and sea levels

The hourly barometric time series data from the meteorological station on Norderney was compared to data from stations "Wittmund" (ca. 39 km from Norderney) on the main land (DWD Climate Data Center (CDC), 2023b) and "Leuchtturm Alte Weser" (ca. 66 km from Norderney) in the German Wadden Sea (DWD Climate Data Center (CDC), 2023a) (Fig. 2b). Figure A1 shows the data of these stations plotted against the data from Norderney and the results of a linear regression analysis performed on these data sets. Data from Norderney and Wittmund are very similar with only little offset, while the data from "Leuchtturm Alte Weser" is offset from data from Norderney by around 6 cm $H_2O$. Cross-correlation analysis shows largest cross-correlations between Norderney station and "Wittmund" at a time lag of 0 h and for "Leuchtturm Alte Weser" at 2 h. Due to the similarities of the data collected at these stations which have spatial differences of tens of kilometers, we assume that spatial variability of barometric pressure at the scale of the study area is negligible.

The sea-level data with 1 min time increments from tide gauge "Norderney Riffgat" was compared to sea-level data from tide gauge "Spiekeroog" (WSV, 2021b) to assess the time shift of the tidal signal to expect along the shoreline of the islands (Fig. 2b). The tide gauge on Spiekeroog is located approximately 35 km east of the tide gauge on Norderney (Fig. 2b) and should thus lag behind the time series observed on Norderney (Malcherek, 2010). Cross-correlation analysis of the sea-level

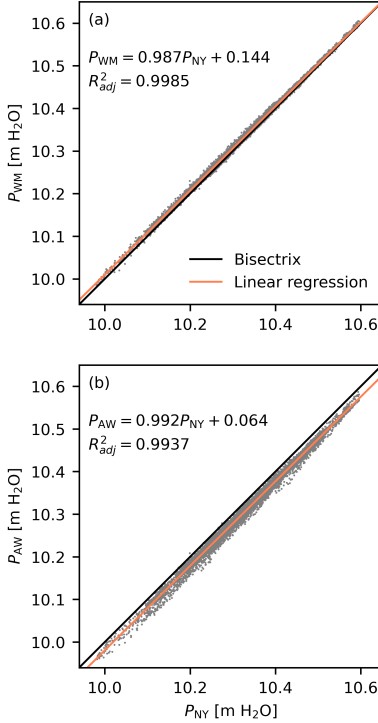

**Figure A1.** Comparison of barometric-pressure ($P$) data collected at the meteorological station on Norderney ($P_{\mathrm{NY}}$) and (a) "Wittmund" ($P_{\mathrm{WM}}$) as well as (b) "Leuchtturm Alte Weser" ($P_{\mathrm{AW}}$) (cf. Fig. 2b). Shown are results of a linear regression analysis performed on the data as well.

time series from Norderney and Spiekeroog shows the maximum correlation at a time lag of 33 min. Thus, the sea-level signal observed on Spiekeroog commonly lags behind the signal observed on Norderney ca. half an hour. Concluding, the temporal offset between tide gauge "Norderney Riffgat" and the shoreline segments close to the groundwater observation wells can expected to be in the order of a few minutes.

## Appendix B: Uncertainty estimation of the response function

The standard error, $\mathrm{SE}_{\mathrm{ORF}}(\tau_k)$, of the ORF at time lag $\tau_k$ is calculated from the $[m^{\mathrm{SL}} \times m^{\mathrm{SL}}]$ covariance matrix $\boldsymbol{\sigma}$ for the instantaneous coefficients $\beta^{\mathrm{SL}}$ obtained by regression deconvolution

$$\mathrm{SE}_{\mathrm{ORF}}(\tau_k) = \sqrt{\sum_{i=0}^{k} \sigma_{ii} + 2\sum_{i=0}^{k}\sum_{j=i}^{k} \sigma_{ij}}, \tag{B1}$$

where $\sigma_{ii}$ is the variance of instantaneous coefficients $\hat{\beta}^{\mathrm{SL}}$ at time lag $\tau_i$, and $\sigma_{ij}$ is the covariance at lags $\tau_i$ and $\tau_j$. The same procedure applies to the BRF.

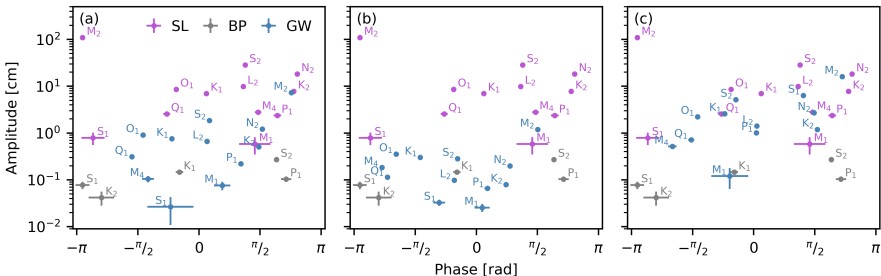

**Figure C1.** Amplitudes and phases obtained using Harmonic Least Squares (HALS) analysis of groundwater-level (GW) time series of monitoring wells (a) BS3, (b) NY-10, and (c) SN12/1. Each plot shows the HALS analysis for sea level (SL) and barometric pressure (BP). Error bars show uncertainty of one standard error.

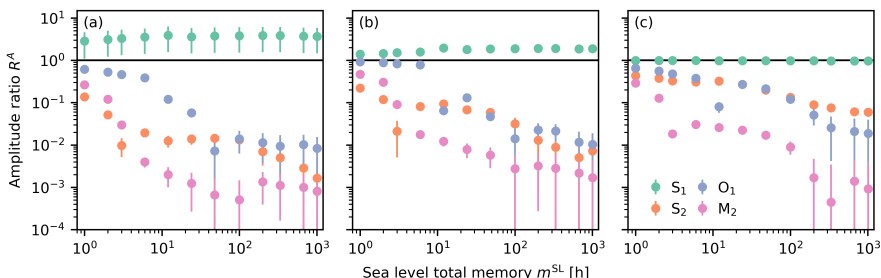

**Figure C2.** Amplitude ratios of observed and corrected groundwater levels (Eq. C1) for tidal constituents obtained by the Harmonic Least Squares (HALS) analysis performed on (a) BS3, (b) NY-10, and (c) SN12/1 as a function of the sea-level memory ($m^{\mathrm{SL}}$). Error bars show uncertainty of one standard error.

## Appendix C: Harmonic least squares analysis of observed and corrected time series

Amplitudes and phases of major tidal constituents (see e.g., McMillan et al., 2019) were obtained from sea-level and groundwater-level time series using Harmonic Least Squares (HALS) analysis (for an outline of HALS see e.g., Schweizer et al., 2021).
Barometric pressure was only analyzed for the subset of tidal constituents relevant to atmospheric tides (Rau et al., 2020). Amplitude and phase uncertainties were estimated as described in Appendix C of Rau et al. (2020).

Results of the HALS analysis are shown in Fig. C1 and identify the semi-diurnal characteristic of the ocean tides with only minor diurnal constituents. This pattern is retained in the groundwater response for BS3 and NY-10 (Fig. C1ab), but the principal diurnal solar constituent $S_1$ is amplified compared to the sea-level signal in the data observed at SN12/1, which
indicates that parts of the spectral power present at this frequency must originate from another process (compare Sect. 3.5).

Figure C2 shows the amplitude ratio

$$R_\nu^A = \frac{A_\nu^{\mathrm{GW_c}}}{A_\nu^{\mathrm{GW}}}. \tag{C1}$$

between the amplitudes of tidal constituent $\nu$ in the observed ($A_\nu^{\mathrm{GW}}$) and corrected ($A_\nu^{\mathrm{GW_c}}$) groundwater time series for different sea-level memories $m^{\mathrm{SL}}$ between 1 h and 6 weeks. Uncertainties are shown as standard errors

$$420 \quad \mathrm{SE}_{R_\nu^A} = |R_\nu^A| \sqrt{\left(\frac{\mathrm{SE}_{A_\nu^{\mathrm{GW_c}}}}{A_\nu^{\mathrm{GW_c}}}\right)^2 + \left(\frac{\mathrm{SE}_{A_\nu^{\mathrm{GW}}}}{A_\nu^{\mathrm{GW}}}\right)^2}, \quad\quad\quad\quad\quad\quad\quad\quad\quad\text{(C2)}$$

obtained by propagating amplitude uncertainties estimated using HALS.

Semi-diurnal constituents, like $M_2$ or $S_2$, are easily removed. A maximum lag of around 6 h suffices for reducing the amplitudes in BS3 and NY-10 below approximately 5 % to 10 % of their original values (Fig. C2ab). However, this is only the case for $M_2$ in SN12/1 (Fig. C2c). Diurnal constituents like $O_1$ require larger total memory of around 12 to 24 h to be reduced equally well (Fig. C2). However, a successful removal of $O_1$ can be assumed for larger amplitude ratios considering the smaller absolute amplitude in the observed signal compared to the semi-diurnal constituents (Fig. C1).

The $S_1$ tidal constituent is not removed from the groundwater signal, and is actually larger in the corrected groundwater signal than in the observed signal in BS3 and NY-10 (Fig. C2ab). Yet, this constituent has little overall effect due to its minor amplitude (Fig. C1ab). As noted in Sect. 3.5, corrected groundwater levels in SN12/1 contain daily signals from nearby production wells. Figure C2c shows that this diurnal pattern maps to $S_1$. The amplification of $S_1$ for BS3 and NY-10 in the corrected time series likely has the same origin and could be caused by removal of an interference between ocean tide's $S_1$ and the daily extraction signal in the observed data.

## Appendix D: Oceanic Response Function at different time series lengths

The *Oceanic Response Function* (ORF) was calculated for smaller portions of the time series from 1 November 2018 to 31 October 2019 to check the dependence of the results on the length of the time series. Analyzed time series lengths were 328.5 days (90 % of the original time series, $n = 2$ samples with different starting points), 292 days (80 %, $n = 3$), 255.5 days (70 %, $n = 4$), 219 days (60 %, $n = 5$), 182.5 days (50 %, $n = 6$), 146 days (40 %, $n = 7$), 109.5 days (30 %, $n = 8$), 73 days (20 %, $n = 9$), and 36.5 days (10 %, $n = 10$). Starting time points of the time series were defined every 36.5 days from 1 November 2018 on. The ORF memory $m^{\mathrm{SL}}$ of 150 h for BS3, 250 h for NY-10, and 48 h for SN12-1 as well as the BRF memory $m^{\mathrm{BP}}$ of 24 h for all monitoring wells was kept unchanged for the analysis. All calculated ORFs are displayed in Figs. S1 to S9 in the supplement.

The maximum value of the ORF depends on the length of the time series and the time frame covered (Fig. D1). Especially for BS3 and NY-10 there also seems to be a dependence on the starting date of the time series, independent of the time series length (Fig. D1ab). For SN12/1, there seems to be a stronger interdependence between starting time point and time series length, where shorter time series that have an earlier starting date show largest $\max(\mathrm{ORF})$ values.

For all three monitoring wells, the $\max(\mathrm{ORF})$ values are generally smaller when a time series only covers the time frame from 1 April 2019 to 31 August 2019, where non-tidal sea-level variability is smaller than in winter and autum (Fig. D2, cf. Fig. 3a). This effect is less pronounced at NY-10 because it is further from the shore and thus the non-tidal sea-level changes have less effect on the groundwater levels at this location (Fig. 3d).

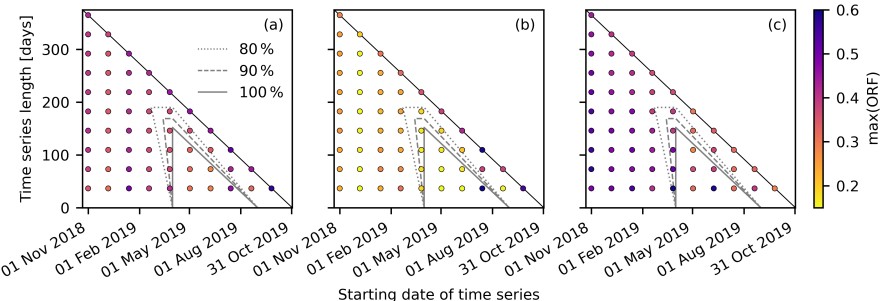

**Figure D1.** Maximum values of the Oceanic Response Function ($\max(\mathrm{ORF})$) as a function of starting date of the time series and time series length for (a) BS3, (b) NY-10, and (c) SN12-1. Contour lines indicate the percentage of a time series within the time frame from 1 April 2019 to 31 August 2019, where the non-tidal sea-level changes are small (cf. Fig. 3a). Only the lower triangle of the plots are filled as in the upper one time series would exceed the end of the studied time frame on 31 October 2019. Bounds of the colorbar are the 5[th] and 95[th] percentile of all $\max(\mathrm{ORF})$ values shown in this figure.

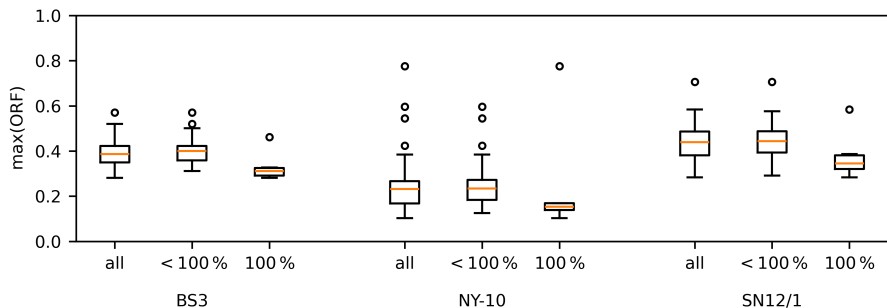

**Figure D2.** Distribution of the maximum values of the Oceanic Response Function ($\max(\mathrm{ORF})$) for the three monitoring wells. Shown are boxplots for all time series shown in Fig. D1 ("all"), for time series which are not entirely within the time frame from 1 April 2019 to 31 August 2019 ("$< 100\,\%$"), and for time series completely within this time frame ("$100\,\%$").

*Code and data availability.* Python scripts and data used in this work are available on Zenodo under https://doi.org/10.5281/zenodo.10868409 (Haehnel and Rau, 2023). An online application (MUFACO: Multi-Factor Correction of Groundwater Levels) to calculate multi-factor regression deconvolution and obtain response functions for multiple stressors is available at https://groundwater.app/app-mufaco/.

*Author contributions.* Patrick Haehnel – formal analysis, data curation, software, visualization, writing - original draft, writing - review & editing; Todd C. Rasmussen – methodology, formal analysis, supervision, writing - original draft, writing - review & editing; Gabriel C. Rau
– conceptualization, methodology, software, supervision, writing - original draft, writing - review & editing.

*Competing interests.*  The authors declare that they have no conflict of interest.

*Financial support.*  Research for this work was funded by the German Federal Ministry of Education and Research (BMBF, Bundesministerium für Bildung und Forschung, project "Water at the Coasts of East Frisia, WAKOS", funding reference number 01LR2003E). Published with the help of the DFG-funded Open Access Publication Fund of the Carl von Ossietzky University Oldenburg.

*Acknowledgements.*  We thank the Stadtwerke Norderney (Municipal Works Norderney), especially O. Rass, for providing groundwater data and support as well as for approving the publication of this data in the Zenodo repository. Thanks to the Wasser- und Schifffahrtsverwaltung des Bundes (German Federal Waterways and Shipping Administration) as well as the Bundesamt für Gewässerschutz (German Federal Institute of Hydrology) for providing tide gauge data. Further, we thank the two reviewers for their constructive comments which helped to significantly improve the paper.

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
