# Peer review of "Technical note: Removing dynamic sea-level influences from groundwater-level measurements"

_Hydrology and Earth System Sciences, 2023_

## Author Response (AR1)

Updated access link to Zenodo repository during peer-review:
https://zenodo.org/records/10161074?token=eyJhbGciOiJIUzUxMiIsImlhdCI6MTcwMTc3NzgzOCwiZXhwI
joxNzA0MzI2Mzk5fQ.eyJpZCI6IjAxN2YzZDYxLWJkMjktNDdhYS1hMjUwLTEyOTUzN2Y5Nzg5YSIsImRhdGEi
Ont9LCJyYW5kb20iOiI5ZDljMmMzMDRlY2VjOWMxYzNmMDYyZDBhOTlhYTlkNiJ9.IJI6huBKnJG31sOHwjO
pXLCnzLAQUtChuZ9h6ao1o0786fdFR9_0vFtMPg2ciP0ldxpt8bujtcvNFKhoUDM9ig

**Response to RC1 from Jonathan Kennel**

This Technical Note introduces the Ocean Response Function which is an application of regression deconvolution using tidal levels as a basis for inputs. It can be used to remove the ocean signal from the water level data. It is well written and presents the underlying study well.

Thank you for your time reviewing the manuscript as well as the constructive and helpful comments provided. Following you find our responses to your comments, which are color coded with blue for neutral, green for agreement, orange for partial agreement, and red for disagreement. Line numbers refer to the revised version of the manuscript unless stated otherwise.

**Moderate concern:**

- In terms of the scientific significance could you describe the difference between the ocean response function and river response function as you see it? Perhaps you might describe how you see them different or similar. Some rivers also have strong tidal effects due to ocean levels or cyclical forcing resulting from dam operations. Does each stressor (barometric, evapotranspiration, pumping, precipitation, Earth tides, river stage, lake levels, ocean tides, anthropogenic loading/unloading, seismic, …) require a separate methods paper when the underlying method is the same but the input varies? Is the key point that for this ocean tide example that the response can be approximated as linear time invariant over the time frame of analysis.

Since we are aware that the method is established within the research community, we decided to communicate our findings as a technical note rather than a full paper. We aim to point out how the complex hydraulic stressor sea level can be approached using regression deconvolution. We were interested in how regression deconvolution would perform under this setting and which time scales would be relevant for the response function to define an adequate system memory. We believe that our findings are worthy of a technical note. The manuscript also provides a generic formulation of the method to include multiple stressors. To our knowledge, present formulations in the literature only cover two simultaneous stressors such as barometric pressure, Earth Tides, and river levels.

We note that the River Response function of Spane and Mackley (2011) focuses on events with periods longer than typical dominating ocean tidal constituents (> 50 h) (Spane and Mackley, 2011; p. 801). Barometric and Earth Tide response functions have a stronger focus on tidal constituents with periods of typically less than 48 h. These differences are also reflected in the system memory, which is much larger in the study of Spane and Mackley (2011) than for Barometric or Earth Tide responses.

Spane and Mackley (2011) argue that if another stressor, like barometric pressure, is to be removed, this should be done in a sequential manner, starting with the most influential stressor. Others (e.g., Toll and Rasmussen, 2007; Rau et al., 2020) have removed two stressors simultaneously and so did we here. We acknowledge that this may have not been addressed well enough in the manuscript and we implemented changes accordingly (see responses to minor comments below).

The method of regression deconvolution is not well recognized in coastal hydrogeology, even though it may be very useful to establish system understanding of coastal aquifers. Research concerned with

obtaining system understanding through the stressor sea level mainly focuses on spectral analysis
(thus frequency-domain focused methods) of groundwater signals in response to ocean tides to
estimate aquifer properties like hydraulic diffusivity from tidal constituents (e.g., Bye and Narayan,
2009; Rotzoll et al. 2008, 2013; many others are outlined by Spane and Mackley, 2011). This approach
typically relies on an analytical solution of Ferris (1952) for tidal propagation in an aquifer and is only
applicable to unconfined aquifers when certain assumptions are fulfilled (observations far from the
shoreline, tidal range in observation well is only small fraction of aquifer thickness, mainly horizontal
flow). For Norderney, none of these requirements are fulfilled for example. Trglavcnik et al. (2018)
additionally used storm flood signals for hydraulic aquifer characterization. In conclusion, the stressors
ocean tides and storm events present in sea-level time series have mostly been analyzed separately.

The time-domain focused regression deconvolution may provide two relevant opportunities for future
research in coastal hydrogeology:

1. Enable hydraulic characterization of coastal aquifers based on regression deconvolution which can
   use the combined information of stresses at all time scales present in the groundwater signal,
   instead of only one (e.g., ocean tides or storm floods), independent of the assumptions required
   by analytical solutions.
2. Allow the extraction of a temporally highly-resolved groundwater recharge signal which is
   typically obscured by the strong sea-level influences at coastal sites near the shoreline. This can be
   relevant as groundwater recharge information, especially at a high temporal and spatial resolution
   is often extremely difficult to obtain.

Therefore, there may not be the need for a technical note for every stressor, but the coastal
hydrogeology community may benefit from the guidance provided in this technical note on how to apply
the method at coastal groundwater observation sites and the further research opportunities this method
potentially provides should it be more recognized in the community.

We agree that the motivation for the technical note should be outlined more clearly and therefore we
revised the objective in the introduction (from Line 32 on):

> "The objective of this work is to (i) provide a generic formulation for regression deconvolution,
> (ii) demonstrate […] unconsolidated sediments, and (iii) illustrate how the method is useful for
> coastal groundwater systems."

The assumption of linearity and time invariance of the response is important but probably hardly ever
fulfilled by field data. As outlined in our responses below (minor comments), the response characteristics
may differ if storm events are present. However, if two time series contain similar sea-level stresses, the
response is comparable in our data set.  Regarding linearity, Spane and Mackley (2011) point out (citing
Smith, 2008; Reilly et al., 1987) that the linear approximation can hold true for a nonlinear system under
certain conditions (changes in saturated aquifer thickness due to the periodic flow are comparably
small). As a coastal aquifer is a nonlinear system regarding sea-level propagation, we added information
regarding this. For Norderney, we assume the linear approximation to be adequate as the saturated
aquifer thickness is likely at around 400-450 m (Haehnel et al., 2023). Following information was added
in Lines 226-236, also including discussions regarding nonlinearities as outline by RC2 for Lines 235-245
in the originally submitted manuscript (see below):

> "While regression deconvolution assumes a linear response of groundwater levels to the
> external influence (cf. Section 2.2), i.e., sea levels, it can be assumed that the response to ocean
> tides is nonlinear due to the changes in aquifer thickness and the low-pass filter effect of the
> aquifer sediment (e.g., Nielsen, 1990; Rotzoll et al., 2008). The latter causes amplitudes of lower-
> frequency tidal constituents to be attenuated less with increasing distance to the shoreline than

higher-frequency ones (Trefry and Bekele, 2004). Further, the phase shift of lower-frequency tidal constituents in the sediment is slower than for higher-frequency ones (Rotzoll et al., 2008). Additionally, higher-harmonic tidal constituents (i.e., shallow water tidal constituents) are generated within the aquifer sediment introducing another source of nonlinearity (Bye and Narayan, 2009).

Smith (2008) reported that linear approximations for periodic flow can be adequate if the changes in saturated aquifer thickness were comparably small, and Reilly et al. (1987) stated a temporally variability of maximum 10 % as a rule of thumb for nonlinear influences. With a tidal range of 2.44 m asl (cf. Section 3.1) and an approximate aquifer thickness of 400 to 450 m (Haehnel et al., 2023), linear approximation seems valid here."

**Minor concerns:**

- Lines 85:88 "It is recommended to perform the deconvolution using the first differences of the measurements, leading to Eq. 5 becoming Δy = β ΔX. This removes the effect of persistent trends in the data and therefore avoids a bias in the regression (Rasmussen and Crawford, 1997; Butler Jr. et al., 2011). " It is also possible to prefilter data or include a background trend term(s) in the regression equation. I would probably not include the difference formulation as a recommendation, but can mention why you did it. In general if appropriate data is available the analysis can be done on non-differenced data. It also avoids the correction required in line 89.

  Thank you for pointing this out. We changed the paragraph's phrasing accordingly (Lines 96-100):

    "The deconvolution was performed using first differences of the measurements, […]. To avoid spurious influences […], the mean of the corrected time series was matched to the uncorrected one."

  The reason for using first differences is outlined in Line 96 (trend removal).

- The term "corrected" is used throughout. While this is commonly used and has been defined before, it suggests that the raw water level is in error. I prefer, the water level with the ____ component(s) removed.

  While we agree with the possible perception of "corrected" as stating the original time series was erroneous, the alternative formulations using "removed" are often cumbersome and prevent concise formulations. To address this and clarify the use of the term "corrected", we added in Lines 93-95:

    "The term "corrected" is used in this work and in the literature regarding regression deconvolution in the sense of "the influence of a process on the time series was removed". The use of the term "corrected" does not suggest any kind of error in the original time series."

- With different length maximum lags you are comparing response functions based on different time periods, you may want to say this or highlight the applicable analysis length. It might not be important with long datasets and relatively short lags, but in shorter length datasets it may be important. Also if the relationship isn't strictly LTI it can be an issue.

  Please refer to our response to the comment below regarding the reproducibility of the ORF for smaller portions of the time series.

130 • Perhaps mention how ocean tides and barometric pressure are spatially variable. What is the influence of the tide monitoring location and weather monitoring location. Is it important for this study?

The spatial variability should not be a concern for this study, but we agree that this should be mentioned.

135 Barometric pressure observed at Norderney is very similar to such observed at other meteorological stations nearby in absolute values and dynamics. We therefore compared the data from Norderney with the data from the two closest meteorological stations "Wittmund" (located on the main land, ca. 39 km south of Norderney) and "Leuchtturm Alte Weser" (located in the German Wadden Sea, ca. 66 km west of Norderney). Cross-correlation analysis of these datasets show little temporal offsets to

140 Norderney (maximum correlation at time lags $\leq 2$ h). Linear regression analysis showed barometric pressure has no discernable offset from Norderney at station "Wittmund" and ca. 6 cm H2O at station "Leuchtturm Alte Weser". Given that these differences occur at a scale of tens of kilometers, the spatial difference of ca. 1.5 km between the meteorological station and the observation wells should not be relevant.

145 We updated Figure 2 in the manuscript to additionally show the location of the two aforementioned meteorological stations and add the following text following Line 181:

"The spatial distance of ca. 1 and 2.5 km between the meteorological station and the groundwater observation wells should not affect the results of this study as the barometric pressure typically varies at larger spatial scales (cf. Appendix A)."

150 For the sea-level data, we compared the data from tide gauge "Norderney Riffgat" with tide gauge "Spiekeroog" located ca. 35 km east of Norderney. Cross-correlation shows that the maximum correlation is found at a time lag of 33 min so that the time difference between the shore segments closest to the observation wells and the tide gauge should be in the order of minutes (well below the measurement time interval).

155 Figure 2 in the manuscript was updated to additionally show the location of tide gauge "Spiekeroog" and the following text was added after Line 188:

"The spatial distance of the tide gauge from the shoreline segments closest to the observation wells should not affect the results presented here, because the temporal offset of the sea-level signal at these shoreline segments compared to the tide gauge is in the order of a few minutes,
160 much shorter than the sampling interval of 1 h used in this study (cf. Appendix A)."

We added Appendix A containing further details on the topic (Lines 346-363):

"**Appendix A: Spatial variability of barometric pressure and sea levels**

The hourly barometric time series data from the meteorological station on Norderney was compared to data from stations "Wittmund" (ca. 39 km from Norderney) on the main land (DWD
165 Climate Data Center (CDC), 2023b) and "Leuchtturm Alte Weser" (ca. 66 km from Norderney) in the German Wadden Sea (DWD Climate Data Center (CDC), 2023a) (Figure 2b). Figure A1 shows the data of these stations plotted against the data from Norderney and the results of a linear regression analysis performed on these data sets. Data from Norderney and Wittmund are very similar with only little offset, while the data from "Leuchtturm Alte Weser" is offset from data
170 from Norderney by around 6 cm $H_2O$. Cross-correlation analysis shows largest cross-correlations between Norderney station and "Wittmund" at a time lag of 0 h and for "Leuchtturm Alte Weser" at 2 h. Due to the similarities of the data collected at these stations which have spatial

differences of tens of kilometers, we assume that spatial variability of barometric pressure at the scale of the study area is negligible.

[Figure]

**Figure A1.** Comparison of barometric-pressure ($P$) data collected at the meteorological station on Norderney ($P_{NY}$) and (a) "Wittmund" ($P_{WM}$) as well as (b) "Leuchtturm Alte Weser" ($P_{AW}$) (cf. Figure 2b). Shown are results of a linear regression analysis performed on the data as well.

The sea-level data with 1 min time increments from tide gauge "Norderney Riffgat" was compared to sea-level data from tide gauge "Spiekeroog" (WSV, 2021b) to assess the time shift of the tidal signal to expect along the shoreline of the islands (Fig. 2b). The tide gauge on Spiekeroog is located approximately 35 km east of the tide gauge on Norderney (Fig. 2b) and should thus lag behind the time series observed on Norderney (Malcherek, 2010). Cross-correlation analysis of the sea-level time series from Norderney and Spiekeroog shows the maximum correlation at a time lag of 33 min. Thus, the sea-level signal observed on Spiekeroog commonly lags behind the signal observed on Norderney ca. half an hour. Concluding, the temporal offset between tide gauge "Norderney Riffgat" and the shoreline segments close to the groundwater observation wells can expected to be in the order of a few minutes."

All other Appendices and references to them were renumbered accordingly throughout the manuscript.

175

180

185

190

- 162:163 "Tidal data were downsampled to hourly intervals for subsequent analysis." Readers may be interested in how - dropping non-matching values or other decimation procedure.

  The sentence in Lines 186-188 was rephrased to make the procedure clearer:

  "Tidal data were downsampled to hourly intervals for subsequent analysis by discarding observation time points that did not match the sampling times of groundwater and barometric pressure data, which were collected at each full hour."

- 183:184 "The precipitation response of BS3 and NY-10 is discernible in mid-August 2019, where groundwater levels increase despite a lack of change in sea levels." Did you calculate a response function for these? If not why not?

  We opted not to include a precipitation response function in our study due to the complexities involved in quantifying the additional impact of precipitation on groundwater recharge and evapotranspiration. Discussing this aspect in depth would have significantly extended the manuscript, as it would entail delving into the intricate mechanisms by which precipitation translates into recharge, especially in unconfined aquifers with shallow groundwater tables. In such scenarios, both evapotranspiration from the unsaturated zone and the saturated zone are pertinent considerations.

  Our primary focus in this manuscript centers on the removal of sea-level influences from groundwater-level time series. As outlined in the manuscript, this approach allows us to isolate the groundwater recharge signal within the time series, effectively eliminating other external factors. This successful isolation was demonstrated in our study with observation wells BS3 and NY-10.

  It is important to note that the time series, once cleaned of all influences except recharge, can potentially be used to calculate recharge values. However, delving into this aspect would open up an entirely new avenue of investigation beyond the scope of our intended objective. Such an exploration would have extended the manuscript beyond the confines of what is typically expected in a Technical Note published in HESS, which is designed to be concise and focused.

- 187 "Periodic and aperiodic sea-level fluctuations" consider simplifying to "Sea-level fluctuations"

  We would like to keep the formulation because it highlights that both ocean tide influences as well as meteorological/storm related influences can be removed simultaneously.

- How reproducible are the response functions when calculated for different portions of the dataset?

  When using less than a year of the dataset, the response functions are not necessarily reproducible. Since different time scales of the stressor are relevant for the ORF (here ocean tides and storm floods), the dataset should be long enough to contain all these time scales if a time-invariant response is desired. For Norderney, the storm events occur almost exclusively during the winter season. Should the dataset not cover such a period, the ORF might look different and the maximum time lag required to remove the influence of sea levels may be shorter as mostly ocean tides are relevant then. We found that time series portions without storm floods present yield smaller maximum ORF values. This is not a shortcoming of the method, but actually highlights that the ORF responds to the stresses present in the sea-level time series.

  We added an additional figure showing oceanic response functions for shorter portions of the time series, updated Figure 3, and discussed the results and implications of this in accordance to Brookfield et al. (2017) in Lines 258-272:

  "Besides the distance to coast and aquifer hydraulic properties, the characteristics of the sea-level fluctuations within the analyzed time period are relevant for the shape of the ORF (Brookfield et al., 2017). For Norderney, the most prominent change in sea-level characteristics is

235     the presence of storm floods during the winter half year and the general lack of them during the summer as well as the generally higher variability of non-tidal sea-level components during winter and autumn (trend line in Fig. 3a). Figure 5 shows ORFs calculated for subsets of the one-year time series of 182.5 days and 73 days with different starting dates (time frames covered are indicated in Figure 3ac). The ORFs are relatively close in shape to the ORF of the entire time

240     series, when winter and/or autumn are covered, i.e., they cover either the start or end of the one-year period. When no time frame with pronounced variability in the non-tidal sea-level component is covered, the maximum ORF value tends to be smaller than that of the entire time series (orange line in Fig. 5b, Figs. S1 to S9), which resembles the then weaker influence of sea-level fluctuations on the groundwater levels (see Appendix C for more details).

245     In the case of the 73-day time series starting in Early-June 2019 (orange line in Figure 5b), sea levels show no pronounced variation beside ocean tides (Figure 3a). Accordingly, the instantaneous coefficients start fluctuating around zero earlier than for the other time series (Figure 5b), indicating a shorter memory $m^{SL}$ of around 48 h. In conclusion, the ORF seems to be time invariant as long as the stresses covered by the individual time series are comparable.

[Figure]

250     **Figure 5.** Oceanic Response Functions (ORF) for BS3 with a time series length of (a) 182.5 days and (b) 73 days. The black response functions and instantaneous coefficients show the results of the analysis of the complete one-year time series (Figure 4a). Colors indicate the time difference of the starting point of the shorter time series from the starting point of the complete time series. Note that not all analyzed response functions are shown (cf. to Fig. S5 and

255     S8). The starting dates of the analyzed time series are displayed by white stripes in the colorbars. Time frames covered by the time series corresponding to the ORFs shown are displayed in matching colors in Figure 3ac."

Figure numbering throughout the manuscript was updated accordingly.

Appendix D was added to the manuscript to further explain the topic by analyzing the given time series for smaller intervals (Lines 396-412):

260     "**Appendix D: Oceanic Response Function at different time series lengths**

The Oceanic Response Function (ORF) was calculated for smaller portions of the time series from 1 November 2018 to 31 October 2019 to check the dependence of the results on the length of

the time series. Analyzed time series lengths were 328.5 days (90 % of the original time series, n = 2 samples with different starting points), 292 days (80 %, n = 3), 255.5 days (70 %, n = 4), 219 days (60 %, n = 5), 182.5 days (50 %, n = 6), 146 days (40 %, n = 7), 109.5 days (30 %, n = 8), 73 days (20 %, n = 9), and 36.5 days (10 %, n = 10). Starting time points of the time series were defined every 36.5 days from 1 November 2018 on. The ORF memory $m^{SL}$ of 150 h for BS3, 250 h for NY-10, and 48 h for SN12-1 as well as the BRF memory $m^{BP}$ of 24 h for all monitoring wells was kept unchanged for the analysis. All calculated ORFs are displayed in Figs. S1 to S9.

The maximum value of the ORF depends on the length of the time series and the time frame covered (Fig. D1). Especially for BS3 and NY-10 there also seems to be a dependence on the starting date of the time series, independent of the time series length (Fig. D1ab). For SN12/1, there seems to be a stronger interdependence between starting time point and time series length, where shorter time series that have an earlier starting date show largest max(ORF) values.

[Figure]

**Figure D1.** Maximum values of the Oceanic Response Function (max(ORF)) as a function of starting date of the time series and time series length for (a) BS3, (b) NY-10, and (c) SN12-1. Contour lines indicate the percentage of a time series within the time frame from 1 April 2019 to 31 August 2019, where the non-tidal sea-level changes are small (cf. Fig. 3a). Only the lower triangle of the plots are filled as in the upper one time series would exceed the end of the studied time frame on 31 October 2019. Bounds of the colorbar are the 5th and 95th percentile of all max(ORF) values shown in this figure.

For all three monitoring wells, the max(ORF) values are generally smaller when a time series only covers the time frame from 1 April 2019 to 31 August 2019, where non-tidal sea-level variability is smaller than in winter and autumn (Fig. D2, cf. Fig. 3a). This effect is less pronounced at NY-10 because it is further from the shore and thus the non-tidal sea-level changes have less effect on the groundwater levels at this location (Fig. 3d).

[Figure]

290    **Figure D2.** Distribution of the maximum values of the Oceanic Response Function (max(ORF)) for the three monitoring wells. Shown are boxplots for all time series shown in Fig. D1 ("all"), for time series which are not entirely within the time frame from 1 April 2019 to 31 August 2019 ("< 100 %"), and for time series completely within this time frame ("100 %")."

We included Figures showing the ORFs of all calculated partial time series from Appendix D as Supplementary Material.

295    The following information was added to the conclusions section from Line 337 on:

"The ORF shape depends on the stresses being present in the sea-level time series and are only similar for different time frames, i.e., time invariant, when the stresses of the time frames are similar. A time frame containing storm events may yield a different ORF than a time frame were ocean tides are the most prominent sea-level influence."

300  • The ocean tide data also includes a barometric related component which may be worth mentioning more directly. This could influence the analysis when barometric pressure and ocean tides were used as it now ends up having correlated data.

Agreed. This cross-correlation existing between the sea-level signal and the barometric pressure signal violates the assumption that inputs are independent of each other (Lines 73-74). The
305    barometric pressure influence is present in the groundwater time series 1) by direct vertical influence, with possibly smaller time lags, and 2) by indirect horizontal influence via the sea levels, with possibly larger time lags. One option would be to perform a regression deconvolution on the sea-level data first using the barometric pressure time series. However, the barometric pressure influence carried by the sea level signal would still be present in the groundwater-level time series
310    and may be erroneously mapped to the barometric response function then.

The barometric response functions at the observation wells have rather small values, so that we expect the vertical influence of the barometric pressure to be much smaller than the indirect horizontal influence through the sea-level signal.

We added a paragraph to outline this in Section 2.1 (Lines 53-57) and updated Figure 1 to show that
315    also sea levels are influenced by barometric-pressure changes:

"Note, that changes in barometric pressure also affect sea level (Boon, 2011), so that barometric influence is introduced into groundwater-level time series of coastal aquifers in two principal directions: (i) vertically, through the direct influence of changes in barometric pressure, and (ii) horizontally, through the indirect influence of barometric pressure on the sea level, which is
320    carried through the aquifer with the propagating ocean tide signal (Figure 1). Hence, the barometric influence affects groundwater levels at different time lags from the vertical and horizontal component, respectively."

The caption of Figure 1 was supplemented with:

"Dotted-grey lines indicate the indirect influence of barometric pressure through sea levels on
325    the groundwater levels."

Further changed/added were:

• **Lines 216-217:** Reformulated, "Periodic and aperiodic sea-level as well as barometric-pressure fluctuations were removed from groundwater-level measurements […]"
• **Lines 188-189 in the originally submitted manuscript:** Deleted "The simultaneous removal
330    of barometric-pressure influences was tested as well, but instantaneous coefficients $\hat{\beta}^{BP}$ were insignificant."

- **Line 189-190 in the originally submitted manuscript:** Moved "This is consistent with shallow water tables and high air permeabilities in the sandy surficial deposits that promotes rapid equilibration of aquifer heads (cf. Table 1 and Fig. 3)." to the paragraph starting in **Line 277** (see below)**.**
- **Line 243:** Change heading to "3.4 Response Functions"
- **Lines 254,282:** Updated the slightly changed values of maximum ORF.
- **Line 277:** Added a paragraph, "The *Barometric Response Function*s (BRF) for BS3 and NY-10 deliver small values and instantaneous coefficients start fluctuating around zero for $\tau > 0$ h for BS3 and $\tau > 4$ h for NY-10 (Fig. 6ab). Thus, the response to barometric-pressure changes is instantaneous (smaller than the measurement interval, BS3) or relatively fast (NY-10). This is consistent with shallow water tables and high air permeabilities in the sandy surficial deposits that promotes rapid equilibration of aquifer heads (cf. average depth to water table in Table 1) (Rasmussen and Crawford, 1997)."
- **Table 1** was supplemented by two additional rows "Average groundwater table [m asl]" and "Average depth to water table [m]" to provide a quantitative measure for the statement above. A new figure (Figure 6) was added to the manuscript, showing the BRF as a function of time lags, to state the barometric influence clearer in general, instead of only stating that direct barometric influence on the groundwater levels was very small.

[Figure]

**Figure 6** Barometric Response Function (BRF) for (a) BS3, (b) NY-10, and (c) SN12/1 with with corresponding instantaneous coefficients $\hat{\beta}^{\mathrm{BP}}$. Vertical error bars indicate an uncertainty of one standard error for the Barometric Response Function (Appendix B).

- **Line 287:** Added "Further, the BRF of SN12/1 shows a pronounced periodic pattern at ca. 2 cpd."
- **Line 317-322:** Added paragraph, "The oscillating BRF of SN12/1 (Figure 6c) is likely a result of the groundwater extraction signal not being present in the regression deconvolution. A

360 similar pattern was observed by Patton et al. (2021) in their analysis of barometric-pressure and Earth-Tide response of groundwater levels in a coastal aquifer regarding ocean tides (they termed this shape "peaked"). In their study, they did not consider sea-level fluctuations and the semi-diurnal ocean tide pattern mapped to the BRF. The oscillation in the BRF of SN12/1 is likely mixed semi-diurnal/diurnal because the tidal constituents $S_1$ and $S_2$ introduced by groundwater extraction are not removed at given memory $m^{SL} = 48\,\text{h}$
365 (Figure C2c)."

- In the ocean response functions, did you compensate for salt water density? Might be worth mentioning if you did or didn't and how it might affect the response function numbers.

No, we did not compensate for salt water density. The variable-density effects are commonly considered negligible when concerned with propagation of ocean tide signals in coastal aquifers
370 (Ataie-Ashtiani et al. 2001; Slooten et al., 2010) and most studies presenting analytical solutions for ocean tide propagation do not consider density effects (e.g., Nielsen 1990; Li et al., 2000; Rotzoll et al., 2008).

**Regarding sea levels:** Calculating freshwater heads for the sea-level time series would change the magnitude of the signal but not the timing. The associated error would be in the order of a few
375 percent of the tidal amplitude, likely substantially less than other uncertainties. Further, the entry point of the seawater into the aquifer is not a discrete point but spread over the sub- and intertidal zone most of the time. Thus, in the current approach, some kind of average elevation for the aquifer-ocean connection would have to be defined. This would offset the absolute values of sea level by a constant factor and would thus be eliminated again when long-running trends are removed by first
380 differencing or another kind of trend removal prior to the analysis.

**Regarding groundwater hydraulic heads:** All observation wells presented in this study are located within the freshwater lens of Norderney and thus measured heads are freshwater heads.

Overall, we do not see any potential effect of salt water density on the response function but agree that the arguments presented above should be briefly mentioned in the manuscript. We included the
385 following information:

- **Lines 175-176: "**All three observation wells are screened entirely in the freshwater lens of the island."
- **Lines 135-138:** Added paragraph, "Note that there is a density difference between seawater and freshwater when applying Eq. (14) with sea levels present in $\Delta X$. Density correction of
390 hydraulic heads is typically achieved by calculating so-called freshwater heads (Post et al., 2007). Correcting measured sea-level data this way would result in a constant offset from the original time series which would be eliminated by the trend removal via first differencing of $X$. Therefore, such a correction was not applied here."

- A key aspect of this is the linear time invariant assumption. I think you should mention this in the
395 applicability portion of the conclusions.

Regarding the time-invariant assumption, the text added to the conclusion regarding the different portions of the time series analyzed (Line 337-341) mentions this topic (see comment above regarding reproducibility of the time series).

Regarding the linearity assumption, we added the following (Lines 339-341):

400 "In the case of Norderney, the assumption of a linear response of groundwater levels to sea-level influences will likely be valid approximately resulting from the small changes in saturated aquifer thickness introduced by the sea-level fluctuations (Reilly, 1987; Smith, 2008)."

- There is no interpretation of the responses. Given that the ocean tides can be considered similar to other surface water bodies - it is likely very similar to a river response function analysis (Brookfield et al, 2017). Interpreting temporal variations in river response functions: an example from the Arkansas River, Kansas, USA

Thank you for the literature suggestion. In the manuscript, we state the reasons for the different responses to sea levels at the different monitoring wells in Lines 253-257. Further interpretations were added according to our response to the comment about reproducibility of the ORF for shorter portions of the time series.

Considering Brookfield et al. (2017), we also added the information that the oceanic response function can change with time as a result of the variable characteristics of the sea-level influence (Lines 246-248):

"Also note that each well has a unique ORF which can also vary with time as a result of temporally variable characteristics of the sea-level influence (Brookfield et al., 2017)."

**Figure suggestions:**

Figure 4: Were ocean levels converted to freshwater head for the comparison? If the goal of this figure is to highlight the signal with the ocean response removed, you may want to make this the focus by using a smaller vertical range for these. Right now they are somewhat obscured by the large y-axis range. I'm not sure it is necessary to repeat the ocean levels and precipitation on each facet and I would probably just have them in separate facets.  This also improves readability by not having dual axes. A subset of the total time can also be helpful.

No, the sea levels were not converted to freshwater heads (cf. to our response above).

We agree that the outline of Figure 4 is not optimal as is. We therefore merged Figures 3 and 4 into a single Figure showing the stressors as in Figure 3ab and all groundwater-level time series shown in Figure 3c and Figure 4 in a separate panel for each monitoring well. We did not include additional subsets of the time series as we believe the major arguments can be followed from the updated figure and this technical note already has many figures.

The caption of the adapted Figure 3 was adjusted to the changed content. And the figure numbering was updated throughout the manuscript.

[Figure]

**Figure 3.** Time series of (a) sea level, (b) barometric pressure and cumulative daily precipitation, and observed as well as corrected ground-water levels in (c) BS3, (d) NY-10, and (e) SN12/1. Oceanic Response Function memories $m^{SL}$ are 150, 250, and 48 h, respectively, while the Barometric Response Function memory $m^{BP}$ is 24 h for each monitoring well. Trends of sea and groundwater levels are shown as well in (a) and (c-e). Colored, horizontal bars in (a) and (c) indicate the time frames covered by shorter portions of the time series for which the ORF was calculated (Fig. 5).

435

Figure 5: Perhaps you want to comment on the oscillations in the response function – what frequency
440   and the potential causes – method related, noise related.

The oscillations visible are at a frequency of approx. 4 cpd for BS3 and NY-10 and at variable between roughly 2.67 and 2 cpd for SN12/1. A spectral analysis (FFT and HALS) was performed on the groundwater-level time series with sea-level and barometric pressure influences removed. This analysis shows that the main frequencies that remain in the groundwater-level signal match the S1 and M4 tidal
445   constituents. Further, frequencies around 2 cpd are present as well.

The S1 constituent originates from the pumping pattern of the production wells as shown in the manuscript for SN12/1, where it is very prominent. Since the island is small the pumping pattern is present in the groundwater-level time series throughout the island but at a much smaller amplitude at BS3 and NY-10.

450   The shallow water tidal constituent M4 likely originates from the generation of higher harmonics of the major tidal constituents in the sediment of the aquifer (Bye and Narayan, 2009). The amplitude of the generated constituent depends on the amplitude of its base frequency and the phase shift likewise on the phase shift of its base frequency (Bye and Narayan, 2009). Since M2 is the dominating tidal constituent at the study site, its first higher harmonic M4 is very likely to be generated in the aquifer and
455   thus present in the time series analyzed here. Since this generated M4 is not present in the sea-level time series it will not be removed by the regression deconvolution, just like the influence of pumping.

Comparing FFT and HALS results shows that the frequencies around 2 cpd do not exactly match the major semi-diurnal tidal constituents. Thus, we assume this to be noise.

This additional information is potentially interesting to readers, therefore we changed the heading of
460   section 3.5 to "Revealing groundwater extraction and aquifer-generated tidal constituents" and added the above mentioned information to this section (Lines 298-316):

"While the pattern of groundwater extraction is clearly visible in the groundwater-level time series of SN12/1, this influence is also present at monitoring wells BS3 and NY-10. To show this, amplitudes of frequencies between 0 and 12 cpd were extracted from the corrected
465   groundwater-level time series using HALS analysis (cf. Appendix C) and the Fast Fourier Transform (FFT) with a Hanning window (Figure 8). This shows the daily groundwater extraction pattern strongly enhances the S1 tidal constituent at SN12/1 to around 6 cm (Fig. 8c) compared to an amplitude of around 0.8 cm present in the ocean-tide signal (Fig. C1). For BS3 and NY-10, the amplitude of tidal constituent $S_1$ introduced by groundwater extraction is much smaller due
470   to the larger distance to the production wells (Figure 8ab).

The groundwater extraction signal is one of the causes of the small amplitude, high-frequency oscillations remaining in the corrected groundwater level time series (Figure 3cd) and the oscillation visible in the instantaneous coefficients of the regression deconvolution (at ca. 4 cpd, cf. Figure 4ab) of BS3 and NY-10. The second source of the oscillations is the generation of the
475   shallow water tidal constituent $M_4$ within the aquifer as a result of the propagation of the tidal signal in the sediment (Bye and Narayan, 2009, cf. Section 3.3). It is generated as the higher harmonic of the $M_2$ constituent, which is the dominating tidal constituent at the study site (Fig. C1), so that the amplitude and phase lag of the generated $M_4$ constituent depend on the amplitude and phase lag of the ocean tide $M_2$ constituent (Bye and Narayan, 2009). For the large
480   amplitude of the ocean tide $M_2$ constituent (Figure B1), the amplitude of the generated $M_4$ constituent is still discernible from noise in the data at the monitoring wells (Figure 8). Further, there is noise present in the corrected time series for frequencies between 0.5 and 3 cpd which cannot be attributed to major tidal constituents (Figure 8), but parts of it may be attributed to

485 the frequency-dependent amplitude attenuation and phase shift of the different tidal constituents within the aquifer sediment (cf. Section 3.3)."

We included an additional figure (Figure 8) showing results of the spectral analysis (FFT and HALS) of the corrected groundwater-level time series (amplitude over frequency) to outline the presence of S1 and M4 in these time series.

[Figure]

**Figure 8.** Amplitudes found in the corrected groundwater-level time series for frequencies between 0.5 and 4.5 cpd obtained with Harmonic Least Squares (HALS) analysis and Fast Fourier Transform (FFT) for (a) BS3, (b) NY-10, and (c) SN12/1. The HALS data shows tidal constituents as outlined in Fig. C1. Note the different y-axis scales on each panel.

Figure 6: I don't think I would highlight the pumping times with the grey boxes in A. It makes it seem like this is actual data. If you include it, I would clearly annotate on the figure to say inferred.

The grey boxes were deleted from Figure 6 and the figure caption was updated accordingly.

**HESS questions**

500 - Does the paper address relevant scientific questions within the scope of HESS? Yes

  - Does the paper present novel concepts, ideas, tools, or data? The tools, concepts, and ideas are well developed previously, data and the relationships are new for this site.

  - Are substantial conclusions reached? Yes/No

  - Are the scientific methods and assumptions valid and clearly outlined? Yes

505 - Are the results sufficient to support the interpretations and conclusions? Yes

  - Is the description of experiments and calculations sufficiently complete and precise to allow their reproduction by fellow scientists (traceability of results)? Yes

  - Do the authors give proper credit to related work and clearly indicate their own new/original contribution? Yes

510 - Does the title clearly reflect the contents of the paper? Yes

  - Does the abstract provide a concise and complete summary? Yes

  - Is the overall presentation well structured and clear? Yes

  - Is the language fluent and precise? Yes

  - Are mathematical formulae, symbols, abbreviations, and units correctly defined and used? Yes

515 - Should any parts of the paper (text, formulae, figures, tables) be clarified, reduced, combined, or eliminated? See comments

  - Are the number and quality of references appropriate? Yes

  - Is the amount and quality of supplementary material appropriate? The supplementary data and code are good. I don't know that the appendix B is necessary.

520  We would like to keep Appendix B (now Appendix C) especially when keeping in mind the community of coastal hydrogeologists. Appendix B shows in detail at which time lags ocean tides would be removed, further elaborating on the statement in Lines 255-257. This gives guidance regarding system memory when aperiodic events are less common at a study site or if the studied time frame does not include such events. While this may seem obvious when familiar with regression deconvolution, we think it may be a
525  helpful addition to the main points of the manuscript for coastal hydrogeologists. Further, Appendix B is now even more referred to in the manuscript following the changes applied in response to the comments of this review.

**Response to RC2 from Rachel Housego**

This manuscript applies a linear regression deconvolution to remove ocean-driven water fluctuations from groundwater heads observed at different three different locations and depths across the barrier island. After removing the ocean-driven forcing the residual groundwater fluctuations are used to understand recharge and pumping responses on the island. Overall, I think the paper is well-written, the figures are easy to interpret and it was interesting to see how well the timing of the fluctuations in the corrected time series did coincide with the pumping schedule. I think some of the methods and conclusions would benefit from adding some additional context about what is unique about applying these functions in a coastal setting to further differentiate it from other papers manuscripts that have applied similar techniques. With some additional clarification I think this manuscript will make a nice contribution to HESS. See specific comments below.

Thank you for your time reviewing the manuscript as well as the constructive and helpful comments provided. In the following, you find our responses to your comments, which are color coded with blue for neutral, green for agreement, orange for partial agreement, and red for disagreement.

A lot of work (although not yet applied in coastal settings) has been done using transfer function noise models, which is also a convolution-based method for groundwater time series analysis but presumes a fixed shape of the response function. I think it would be worth citing some of this work and noting the difference in the methods.

Collenteur, R. A., Bakker, M., Caljé, R., Klop, S. A., & Schaars, F. (2019). Pastas: open source software for the analysis of groundwater time series. Groundwater, 57(6), 877-885.

We added an explanation of the difference between the regression deconvolution method and transfer function noise (TFN) modeling and include appropriate references. It points out that regression deconvolution does not rely on the a-priori assumption of the response function (e.g., Gamma distribution), but rather estimates the response function directly from the data, i.e., the shape is not predefined by a chosen response function model, which is of advantage for gaining process understanding.

The added paragraph is found in Lines 139-144:

"Besides regression deconvolution, transfer function noise models are used to model groundwater-level time series from time series of stresses (e.g., groundwater recharge, groundwater extraction, sea levels) using convolution (e.g., von Asmuth et al., 2002; Collenteur et al., 2019; Bakker and Schaars, 2019) and to estimate unknown stresses from groundwater-level time series (e.g., Collenteur et al., 2021; Pezij et al., 2020). The method differs from regression deconvolution in that the response function is pre-defined with a fixed shape, typically by a probability density function like the Gamma distribution (Collenteur et al., 2019), and not obtained through the data itself."

Neglecting wave set-up likely causes an issue in removing the oceanic effects on water levels, especially during surges. For more see the following papers and references therein.

da Silva, P. G., Coco, G., Garnier, R., & Klein, A. H. (2020). On the prediction of runup, setup and swash on beaches. Earth-Science Reviews, 204, 103148.

Stockdon, H. F., Holman, R. A., Howd, P. A., & Sallenger Jr, A. H. (2006). Empirical parameterization of setup, swash, and runup. Coastal engineering, 53(7), 573-588.

We agree that waves may affect the oceanic response function. However, due to the high frequency of wave action, these effects would have very little to no effect on the data observed at the monitoring wells in this study. This is due to the low-pass filtering of the sediment which cancels the influence of high-frequency wave action over propagation distance. Wave setup can contribute significantly to groundwater table overheight also induced by tidal motion (Nielsen, 1990; 1999), but the oceanic response function focusses on the time-series dynamics rather than more persistent offsets such as is caused by wave-induced overheight.

We included an explanation with a recommendation to include wave-setup data when analyzing groundwater-level time series close to the shoreline (Lines 237-242):

"Besides ocean tides, waves can have a pronounced impact on near-shore groundwater-level dynamics (e.g., Nielsen, 1999; Housego et al., 2021). Due to the generally high-frequency of the wave dynamics at the shoreline (e.g., Stockdon et al., 2006; Hegge and Masselink, 1991) and the low-pass filter properties of the aquifer sediment (e.g., Rotzoll et al., 2008; Trefry and Bekele, 2004), waves can be assumed not to impact the groundwater-level dynamics at the monitoring wells in this study, which are several hundreds of meters from the shoreline (cf. Table 1). However, the influence of wave dynamics on groundwater levels may be relevant at beach sites or sites closer to the shoreline."

How does the ORF behave in the frequency domain? Is it consistent with what is known about how ocean driven water table fluctuations propagate through the subsurface, e.g. longer wave periods propagate faster and attenuate less?

We note that ORFs are not comparable with frequency domain analysis. With the given memory of the ORF (48 h, 150 h, and 250 h), the ORFs are too short for a reliable frequency domain analysis, like Fast Fourier Transform, as spectral leakage would prevent meaningful interpretation. Figure B2 in the manuscript shows that different time lags are required to remove different tidal constituents successfully; e.g., the diurnal constituent O1 is removed less successfully than the semi-diurnal M2 (amplitude ratio of O1 is larger than that of M2 at all memories). This is related to the differing propagation properties of the tidal constituents, where O1 is less attenuated than M2. Since this non-linearity is introduced within the aquifer, it cannot be fully removed from the groundwater-level time series using regression deconvolution. As mentioned further below (in response to the comment on Lines 235-245 in the originally submitted manuscript), we added this information in the discussion of the results.

Also in response to a comment of RC1 regarding Figure 5, we added an additional Figure showing the amplitudes in the frequency domain of the corrected time series (Figure 8). This shows that all major tidal constituents but S1 and the shallow water constituent M4 are successfully removed from the groundwater-level time series.

605  How sensitive are the ORF parameters to the storm? There is only one in the data set, if it is removed how different is the maximum time lag at each well? I think it may be possible that the maximum time lag is heavily influenced by the intensity and duration of the surges in the data set and if a more extreme event was measured that parameter may be different.

610  We agree that the presence of storm events is relevant, and we included more information about this in the revised manuscript. Please refer to our response to RC (comment "How reproducible are the response functions when calculated for different portions of the dataset?"). We added a figure (Figure 5) and additional explanation pointing out the relevance of storm events for the shape of the ORF (Lines 258-272).

615  Would there be any benefit to applying the tidal constituents and the moving-averaged trend as separate drivers/would the residual be different if you took that approach?

620  This could be beneficial if one was only interested in the response to storm events for example. Then extracting a trend time series using a filter designed to remove ocean tides from time series could make sense. This trend time series could then be analyzed using regression deconvolution to obtain the response to storm events and other non-tidal sea-level influences. The residuals would look different if only a single sea-level influence (i.e., ocean tides or storm events) were removed because one of the drivers would remain in the signal. Separating the drivers could generally make sense depending on the question at hand. Since this technical note is concerned with removing the entire sea-level influence, this is beyond the scope of this manuscript.

What concerns are there for overfitting and aliasing with this approach?

625  Regarding overfitting, the concerns would be the same as for linear regression analysis. Thus, the number of parameters (here, time-lagged input time series) should be (substantially) less than the number of data points. In the case of regression deconvolution, "too many" parameters would result in a lot of corresponding regression coefficients fluctuating around zero. This can introduce spurious fluctuations to the corrected time series.

630  Regarding overfitting we added the following paragraph to the manuscript (Lines 273-276):

"Note, that generally the maximum number of time lags (i.e., number of instantaneous coefficients) used in the regression deconvolution should only constitute a small portion of the number of time steps present in the analyzed time series to avoid overfitting. Thus, systems with longer memory require longer time series to produce meaningful response functions. In our case
635  for example, the longest required memory of 250 h is around 3 % of the one-year time series."

Regarding aliasing, the sampling frequency should be such, that all relevant ocean tide constituents can be captured in the observed time series. Since the memory (maximum time lag) depends on the tidal frequencies and storm events present, sampling intervals exceeding a threshold may lead to misleading ORF shapes and imprecise time lag information in general. Due to the strong attenuation of high
640  frequency tidal constituents (i.e., shallow water tidal constituents), the sampling interval of 1 h used in this study should suffice to avoid such issues. Besides, with the respective Nyquist frequency at 12 cpd in this case, the most relevant shallow water tidal constituents would also not cause an aliasing issue in this case.

645    We added a sentence about this to the manuscript (Lines 171-173):

"At the given time series length of one year, time increments of one hour are generally sufficient to capture the tidal constituents present at the study site (Schweizer et al., 2021)."

Would the generated ORF function have the ability to forecast future groundwater levels based on ocean water level time series? Under what environmental conditions would this not work as well?

650    While interesting, this is beyond the scope of this technical note but would be worthwhile investigating in a future contribution. If all relevant sea-level stressors (here: ocean tides and storm events) are present in the time series the ORF is derived from and this time series is long enough to cover the variability of these stressors, it should generally be possible to forecast groundwater levels. As other forecasting methods that rely on training data and statistical models, environmental conditions not
655    present in the training data could pose a challenge to the forecast. Thus, should general characteristics of tides and storm events change in the future, forecasts calculated based on regression deconvolution would probably not be reliable.

Line 235-245 I think this is presented as being too generally applicable to all coastal environments however some field studies have shown non-linear responses to ocean forcing, developing skewness and
660    asymmetry in the water table fluctuations that I do not think this method would be able to remove.

E.g. Raubenheimer, B., Guza, R. T., & Elgar, S. (1999). Tidal water table fluctuations in a sandy ocean beach. Water Resources Research, 35(8), 2313-2320.

In addition to the discussion of the linearity assumption as outlined in our response to the moderate concern comment of RC1, we also mentioned the changing shape of the propagating ocean tide signal in
665    the aquifer as a result of the different attenuation and phase shift properties of tidal constituents at different frequencies and due to the generation of higher harmonic tidal constituents in the aquifer sediment (Bye and Narayan, 2009) (paragraphs added in Lines 226-236). This potentially also adds to the noise present in the corrected time series, that cannot be explained from other sources, which is mentioned in addition to discussions introduced following RC1 on Figure 5 in Lines 299-316.

670    We reformulated the paragraph of the conclusions referenced in this comment as follows (Lines 329-336):

"Our findings expand the range of applications for regression deconvolution by enabling the characterization and mitigation of external perturbations impacting groundwater levels. These perturbations encompass barometric pressure, Earth tide, river stage fluctuations, and now,
675    oceanic influences. Our methodology is well-suited for analyzing data obtained from groundwater monitoring in oceanic and coastal aquifers. This capability is instrumental in enhancing our understanding and sustainable management of these critical water systems. Future research endeavors should prioritize a systematic exploration of how hydraulic processes (e.g., modulation of tidal signals within aquifer sediments) and properties (e.g., hydraulic
680    diffusivity) in coastal aquifers affect Oceanic Response Functions. Additionally, estimating response functions linked to groundwater extraction becomes an important area for investigation once suitable data becomes available."

**Further technical corrections**

**Affiliation:** The word "University" in the affiliation of author "Patrick Haehnel" had to be changed to the German word "Universität" according to new guidelines of University of Oldenburg regarding affiliation statements.

**Figure 2:** The country code for Austria was wrong ("AU") and corrected to "AT". A box was added in (b) to outline the spatial extent of (c). In (b), "Wangerooge" was missing the final letter "e" which was included in the updated version. In the figure caption "Haehnel et al. (under review)" was changed to "Haehnel et al. (2023)" as the article is published now. In the caption a misplaced "from" was deleted.

**Figure C1:** Replaced "groundwater" with "groundwater-level" in figure caption.

**Table 1:** Was adapted to fit in a single column. No content changes were made besides the ones mentioned in the responses to reviewer comments. A "the" was added in the caption.

**Equations:** Harmonized use of brackets around equation numbers throughout the manuscript.

**Equation 14/Line 133:** There was a erroneously placed subscript "i" for variable t in "$\Delta X^p(t_i - \tau_k)$" which was removed. "GW" was corrected to non-italic font.

**Lines 12-13:** Added ", saline" to be more precise.

**Line 13-14:** Replaced "these fluids" with "fresh- and saltwater".

**Line 28:** Replaced "groundwater" with "groundwater-level".

**Line 127:** Replaced "characterises" with "characterizes".

**Line 157:** Deleted comma before "and".

**Line 173:** Replaced "these" with "the".

**Line 196:** Added "related to Norderney" to be more precise.

**Line 197:** Replaced "have" with "were converted to" to be more precise.

**Line 198:** Comma after "filter" was removed.

**Line 214:** Added "to early-October 2019" to be more precise.

**Line 244:** Replaced "Ocean" with "Oceanic".

**Line 287:** Added "(cf. Sect. 3.5)".

**Line 326:** Replaced "characterising" with "characterizing".

**Line 342:** Deleted "barometric and" due to changes implemented in the manuscript regarding BRFs.

**Line 368-369:** Added "The same procedure applies to the BRF." due to changes implemented in the manuscript regarding BRFs.

**Lines 371-372:** Replaced "groundwater levels" with "groundwater-level time series".

**Lines 414-415:** Included "An online application (MUFACO: Multi-Factor Correction of Groundwater Levels) to calculate multi-factor regression deconvolution and obtain response functions for multiple stressors is available under https://groundwater.app/app-mufaco/."

**Lines 420-421:** Updated the funding statement to "Research for this work was funded by the German Federal Ministry of Education and Research (BMBF, Bundesministerium für Bildung und Forschung, project "Water at the Coasts of East Frisia, WAKOS", funding reference number 01LR2003E)". No change of the funding source just a more precise mentioning of the funder and the funded project.

**Lines 421-422:** Updated the APC funding sentence to the requirements of the University of Oldenburg: "Published with the help of the DFG-funded Open Access Publication Fund of the Carl von Ossietzky University Oldenburg."

**Lines 430-431:** Corrected the reference for Barlow et al. (2003).

**Lines 448-453:** The reference for hourly pressure and precipitation data from the DWD was split into a separate reference for each variable since we noticed upon checking the references that they were obtained through different URLs and thus constitute two distinct datasets.

**Lines 462-463:** The reference of Falkland (1991) was corrected.

**Lines 472-473:** The reference to the hydrogeological model of Norderney was updated to the now published article.

**Lines 490-491:** Corrected the reference for Li et al. (2004).

**Lines 498-500:** Corrected the reference for NLWKN (2021).

**Lines 549-550:** Corrected the reference for Spane (2002).

**Zenodo repository:** Was updated with the additional analyses introduced for the revisions. Reference and DOI was updated accordingly.

**References**

Ataie-Ashtiani, B., Volker, R. E., & Lockington, D. A. (2001). Tidal effects on groundwater dynamics in unconfined aquifers. Hydrological Processes, 15(4), 655–669. https://doi.org/10.1002/hyp.183

Bakker, M., & Schaars, F. (2019). Solving Groundwater Flow Problems with Time Series Analysis: You May Not Even Need Another Model. Groundwater, 57(6), 826–833. https://doi.org/10.1111/gwat.12927

Brookfield, A. E., Stotler, R. L., & Reboulet, E. C. (2017). Interpreting temporal variations in river response functions: an example from the Arkansas River, Kansas, USA. Hydrogeology Journal, 25(5), 1271–1282. https://doi.org/10.1007/s10040-017-1545-9

Bye, J. A. T., & Narayan, K. A. (2009). Groundwater response to the tide in wetlands: Observations from the Gillman Marshes, South Australia. Estuarine, Coastal and Shelf Science, 84(2), 219–226. https://doi.org/10.1016/j.ecss.2009.06.025

Collenteur, R. A., Bakker, M., Caljé, R., Klop, S. A., & Schaars, F. (2019). Pastas: Open Source Software for the Analysis of Groundwater Time Series. Groundwater, 57(6), 877–885. https://doi.org/10.1111/gwat.12925

Collenteur, R. A., Bakker, M., Klammler, G., & Birk, S. (2021). Estimation of groundwater recharge from groundwater levels using nonlinear transfer function noise models and comparison to lysimeter data. Hydrology and Earth System Sciences, 25(5), 2931–2949. https://doi.org/10.5194/hess-25-2931-2021

DWD Climate Data Center (CDC). (2023a). Historical hourly station observations of pressure for Leuchtturm Alte Weser (station 0102) from 1949 to 2022, version v23.3 [Data set]. Retrieved from https://opendata.dwd.de/climate_environment/CDC/observations_germany/climate/hourly/pressure/historical/ [23 August 2023]

DWD Climate Data Center (CDC). (2023b). Historical hourly station observations of pressure for Wittmund (station 5640) from 1949 to 2022, version v23.3 [Data set]. Retrieved from https://opendata.dwd.de/climate_environment/CDC/observations_germany/climate/hourly/pressure/historical/ [23 August 2023]

Falkland, A. (Ed.). (1991). Hydrology and water resources of small islands: a practical guide. Paris: UNESCO.

Haehnel, P., Freund, H., Greskowiak, J., & Massmann, G. (2023). Development of a three-dimensional hydrogeological model for the island of Norderney (Germany) using GemPy. Geoscience Data Journal, 00, 1–17. https://doi.org/10.1002/gdj3.208

Hegge, B. J., & Masselink, G. (1991). Groundwater-Table Responses to Wave Run-up: An Experimental Study From Western Australia. Journal of Coastal Research, 7(3). Retrieved from https://journals.flvc.org/jcr/article/view/78520

Housego, R., Raubenheimer, B., Elgar, S., Cross, S., Legner, C., & Ryan, D. (2021). Coastal flooding generated by ocean wave- and surge-driven groundwater fluctuations on a sandy barrier island. Journal of Hydrology, 603, 126920. https://doi.org/10.1016/j.jhydrol.2021.126920

Li, L., Barry, D. A., Stagnitti, F., Parlange, J.-Y., & Jeng, D.-S. (2000). Beach water table fluctuations due to spring-neap tides: moving boundary effects. Advances in Water Resources, 23(8), 817–824. https://doi.org/10.1016/S0309-1708(00)00017-8

Malcherek, A. (2010). Gezeiten und Wellen, Die Hydromechanik der Küstengewässer. Wiesbaden: Vieweg+Teubner. https://doi.org/10.1007/978-3-8348-9764-0

Nielsen, P. (1990). Tidal dynamics of the water table in beaches. Water Resources Research, 26(9), 2127–2134. https://doi.org/10.1029/WR026i009p02127

Nielsen, P. (1999). Groundwater Dynamics and Salinity in Coastal Barriers. Journal of Coastal Research, 15(3), 732–740.

Patton, A. M., Rau, G. C., Cleall, P. J., & Cuthbert, M. O. (2021). Hydro-geomechanical characterisation of a coastal urban aquifer using multiscalar time and frequency domain groundwater-level responses. Hydrogeology Journal, 29, 2751–2771. https://doi.org/10.1007/s10040-021-02400-5

Pezij, M., Augustijn, D. C. M., Hendriks, D. M. D., & Hulscher, S. J. M. H. (2020). Applying transfer function-noise modelling to characterize soil moisture dynamics: a data-driven approach using remote sensing data. Environmental Modelling & Software, 131, 104756. https://doi.org/10.1016/j.envsoft.2020.104756

Post, V., Kooi, H., & Simmons, C. (2007). Using Hydraulic Head Measurements in Variable-Density Ground Water Flow Analyses. Ground Water, 45(6), 664–671. https://doi.org/10.1111/j.1745-6584.2007.00339.x

Rasmussen, T. C., & Crawford, L. A. (1997). Identifying and Removing Barometric Pressure Effects in Confined and Unconfined Aquifers. Groundwater, 35(3), 502–511. https://doi.org/10.1111/j.1745-6584.1997.tb00111.x

Rau, G. C., Cuthbert, M. O., Acworth, R. I., & Blum, P. (2020). Technical note: Disentangling the groundwater response to Earth and atmospheric tides to improve subsurface characterisation. Hydrology and Earth System Sciences, 24(12), 6033–6046. https://doi.org/10.5194/hess-24-6033-2020

Reilly, T. E., Franke, O. L., & Bennett, G. D. (1987). The Principle of Superposition and Its Application In Ground-Water Hydraulics. In Applications in Hydraulics (Vol. Book 3). Washington: U.S. Geological Survey. Retrieved from https://pubs.usgs.gov/twri/twri3-b6/

Rotzoll, K., El-Kadi, A. I., & Gingerich, S. B. (2008). Analysis of an Unconfined Aquifer Subject to Asynchronous Dual-Tide Propagation. Ground Water, 46(2), 239–250. https://doi.org/10.1111/j.1745-6584.2007.00412.x

Rotzoll, K., Gingerich, S. B., Jenson, J. W., & El-Kadi, A. I. (2013). Estimating hydraulic properties from tidal attenuation in the Northern Guam Lens Aquifer, territory of Guam, USA. Hydrogeology Journal, 21(3), 643–654. https://doi.org/10.1007/s10040-012-0949-9

Schweizer, D., Ried, V., Rau, G. C., Tuck, J. E., & Stoica, P. (2021). Comparing Methods and Defining Practical Requirements for Extracting Harmonic Tidal Components from Groundwater Level Measurements. Mathematical Geosciences, 53(6), 1147–1169. https://doi.org/10.1007/s11004-020-09915-9

Slooten, L. J., Carrera, J., Castro, E., & Fernandez-Garcia, D. (2010). A sensitivity analysis of tide-induced head fluctuations in coastal aquifers. Journal of Hydrology, 393(3–4), 370–380. https://doi.org/10.1016/j.jhydrol.2010.08.032

Smith, A. J. (2008). Weakly Nonlinear Approximation of Periodic Flow in Phreatic Aquifers. Groundwater, 46(2), 228–238. https://doi.org/10.1111/j.1745-6584.2007.00418.x

Spane, F. A., & Mackley, R. D. (2011). Removal of River-Stage Fluctuations from Well Response Using Multiple Regression. Groundwater, 49(6), 794–807. https://doi.org/10.1111/j.1745-6584.2010.00780.x

Stockdon, H. F., Holman, R. A., Howd, P. A., & Sallenger, A. H. (2006). Empirical parameterization of setup, swash, and runup. Coastal Engineering, 53(7), 573–588. https://doi.org/10.1016/j.coastaleng.2005.12.005

Toll, N. J., & Rasmussen, T. C. (2007). Removal of Barometric Pressure Effects and Earth Tides from Observed Water Levels. Groundwater, 45(1), 101–105. https://doi.org/10.1111/j.1745-6584.2006.00254.x

Trefry, M. G., & Bekele, E. (2004). Structural characterization of an island aquifer via tidal methods. Water Resources Research, 40(1), W01505. https://doi.org/10.1029/2003WR002003

Trglavcnik, V., Morrow, D., Weber, K. P., Li, L., & Robinson, C. E. (2018). Analysis of Tide and Offshore Storm-Induced Water Table Fluctuations for Structural Characterization of a Coastal Island Aquifer. Water Resources Research, 54(4), 2749–2767. https://doi.org/10.1002/2017WR020975

von Asmuth, J. R., Bierkens, M. F. P., & Maas, K. (2002). Transfer function-noise modeling in continuous time using predefined impulse response functions. Water Resources Research, 38(12), 23-1-23–12. https://doi.org/10.1029/2001WR001136

Wasserstraßen- und Schifffahrtsverwaltung des Bundes (WSV) [Federal Waterways and Shipping Administration]. (2021b). Sea level time series data for tide gauge Spiekeroog from 1999 to 2021 (1 min) [Data set]. Provided by Bundesanstalt für Gewässerkunde (BfG) [German Federal Institute of Hydrology].

---

## Referee Report (RR1)

I appreciate the author's efforts in revising their manuscript in response to reviewer comments and I look forward to seeing this in print at HESS. At this stage there is only one issue about the effect of wave set-up in the author's reply that needs to be resolved before this can be published. Due to the application of this paper specifically to coastal settings I think it is important that this is clarified in the final version of the manuscript. See full response below to the original discussion.

Neglecting wave set-up likely causes an issue in removing the oceanic effects on water levels, especially during surges. For more see the following papers and references therein. 565
da Silva, P. G., Coco, G., Garnier, R., & Klein, A. H. (2020). On the prediction of runup, setup and swash on beaches. Earth-Science Reviews, 204, 103148.
Stockdon, H. F., Holman, R. A., Howd, P. A., & Sallenger Jr, A. H. (2006). Empirical parameterization of setup, swash, and runup. Coastal engineering, 53(7), 573-588.

We agree that waves may affect the oceanic response function. However, due to the high frequency of wave action, these effects would have very little to no effect on the data observed at the monitoring wells in this study. This is due to the low-pass filtering of the sediment which cancels the influence of high-frequency wave action over propagation distance. Wave setup can contribute significantly to groundwater table overheight also induced by tidal motion (Nielsen, 1990; 1999), but the oceanic response function focusses on the time-series dynamics rather than more persistent offsets such as is caused by wave-induced overheight.
We included an explanation with a recommendation to include wave-setup data when analyzing groundwater-level time series close to the shoreline (Lines 237-242):
"Besides ocean tides, waves can have a pronounced impact on near-shore groundwater-level dynamics (e.g., Nielsen, 1999; Housego et al., 2021). Due to the generally high-frequency of the wave dynamics at the shoreline (e.g., Stockdon et al., 2006; Hegge and Masselink, 1991) and the low-pass filter properties of the aquifer sediment (e.g., Rotzoll et al., 2008; Trefry and Bekele, 2004), waves can be assumed not to impact the groundwater-level dynamics at the monitoring wells in this study, which are several hundreds of meters from the shoreline (cf. Table 1). However, the influence of wave dynamics on groundwater levels may be relevant at beach sites or sites closer to the shoreline."

I agree with the authors that the high frequency effects of wave action would not affect the inland groundwater levels. However, the net effect of wave set-up during storms is a long-term modification of the **mean water level at the shoreline due to conservation of momentum from wave breaking** which actually has a long effective wave period and would not be attenuated as described above and therefore could impact inland levels. **It is not a wave-by-wave process.** For example, if there were waves at 5 m offshore for a 2-day period during that entire two-day duration the water level at the shoreline would be elevated 1-1.25 m above the level predicted by using an offshore wave buoy to design the ocean time series. During calm conditions you can neglect this effect but during the storm responses this becomes important. The effect of set-up does attenuate inland and likely becomes less significant beyond 500 m inland where your sites are located. However, this is a methods paper specific to coastal settings so I think it is really important to present this accurately because this could be transferred to other coastal sites where wave setup would be important in terms of designing an accurate ORF, especially for time series where multiple surge events are being removed.

---

## Author Response (AR2)

**Updated access link to Zenodo repository during peer-review:**
https://zenodo.org/records/10868409?token=eyJhbGciOiJIUzUxMiIsImlhdCI6MTcwNjYyNzAyMiwiZXhwIj
oxNzM1NjAzMTk5fQ.eyJpZCI6IjIwYjE3YTU1LWExNTktNDc3NS04NDQwLTdlYmM1NTljMTNjNyIsImRhdGEi
Ont9LCJyYW5kb20iOiIzMGU2ZTBlZTc1Y2JlMzRhOGIzYTkwYjk2NDg2NjM3OCJ9.Cr9YcHfeJ1kOV3cF8Nb5a
meI97Kvnf3YKNVW3Q56qR8wI25tKcZrubEBEYysB7pVFKXGNfNuGOUNIM0dp4r0wA

We thank both reviewers for their time reviewing the revised version of this manuscript as well as for the constructive and helpful comments provided. Following you find our responses to your comments, which are color coded with blue for neutral, green for agreement, orange for partial agreement, and red for disagreement. Line numbers refer to the revised version of the manuscript unless stated otherwise.

**Response to report of referee #1, Jonathan Kennel**

**Lines 135-139:** Previously I commented on this, but I still think it still needs clarification. The current text states this is an offset effect, but I think it is a scaling effect. When comparing sea level and freshwater I would use the same units (i.e. convert sea level to an equivalent freshwater or vice-versa). For example a change of 1 m sea level will be equivalent to a change 1.027 m of freshwater based on its density. This is the driving force for the impulse response and I don't think taking first differences has any effect on this. This directly scales the Ocean Response Function by ~ 2.7% and if you don't adjust for this your maximum ORF will be increased by this amount compared to the freshwater-freshwater comparison. Unless you want to define the Ocean Response Function in terms of salt water to freshwater (which is probably not the way to go) I think this should be changed. Consider the analogy to barometric pressure – you always want the units to be the same for the barometric response function or barometric efficiency calculation. This would affect the figures as well but the overall story of the paper remains the same.

Thank you for pointing this out again. In our revisions, we argued with the measured heads when we should have considered the head differences. We revised the paragraph from Lines 135 to 138 in submitted manuscript accordingly and put the additional explanations in a separate paragraph **(Lines 156-178)**:

"**2.5 Considering density effects**

The density difference between seawater and freshwater has to be considered when applying Eqs. (8), (9), and (14) with sea levels present in $\Delta X$. Here, the ORF is defined based on hydraulic head measurements in freshwater. The propagation of external influences in the aquifer depends on the pressure of the external stressor rather than the elevations, which are used as a proxy (i.e. hydraulic heads). A change of hydraulic head in seawater yields a larger pressure change than the same change in freshwater would due to the density difference. Therefore, sea-level records need to be corrected for this higher density to correctly represent the pressure changes of the sea level at the shore with reference to fresh groundwater inland.

Density correction of hydraulic heads is typically achieved by calculating freshwater heads

$$h_f(t) = \frac{\rho}{\rho_f} h(t) - \frac{\rho - \rho_f}{\rho_f} z, \tag{15}$$

where $h$ is the measured point water head, $\rho_f$ is the freshwater density ($1000 \text{ kg m}^{-3}$) and $\rho$ is the density of the water at the screen elevation $z$ of a monitoring well (Post et al., 2007). In case of sea-level observations, $\rho$ is the seawater density and $z$ is the elevation of the sea floor. When using first differences, the freshwater head difference between times $t_i$ and $t_{i-1}$ is

$$\Delta h_f = h_f(t_i) - h_f(t_{i-1}) = \frac{\rho}{\rho_f} [h(t_i) - h(t_{i-1})] = \frac{\rho}{\rho_f} \Delta h \tag{16}$$

so that sea-level differences in Eqs. (9) and (14) have to be defined as freshwater-equivalent differences

45
$$\Delta X_f^{\mathrm{SL}} = \frac{\rho}{\rho_f} \Delta X^{\mathrm{SL}} \tag{17}$$

which corrects differences from measured sea levels $\Delta X^{\mathrm{SL}}$ by the density ratio $\rho/\rho_f$ between salt- and freshwater.

Should the groundwater monitoring well be screened in a location of brackish water or saltwater, the density correction needs to be applied to the hydraulic head differences as well to
50     obtain freshwater-equivalent hydraulic head differences

$$\Delta Y_f(t) = \frac{\rho(t)}{\rho_f} \Delta Y, \tag{18}$$

which allows to obtain comparable ORFs between monitoring sites. Especially at beach sites, the density ratio may be a function of time reflective of salinity changes around the screen of the monitoring well (Greskowiak and Massmann, 2021; Grünenbaum et al., 2023). Details on the
55     estimation of groundwater density from electric conductivity measurements are provided by Post (2012)."

The paragraph about transfer function noise models in Lines 139 to 144 of the submitted manuscript was kept in Section 2.4, now found following Eq. (14) in **Lines 150-155.**

Site-specific information regarding the density correction was added in **Lines 224 to 225**:

60     "Sea-level differences as required for Eq. (14) were converted to freshwater-equivalent sea-level differences according to Eq. (17) with density ratio $\rho/\rho_f = 1.025$, assuming a saltwater density of 1025 kg m$^{-3}$ at the study site."

Figures showing results regarding the regression deconvolution were adapted according to the conversion of sea-level differences to freshwater-equivalent sea-level differences (Figs. 3, 4, 5, 6, 7, 8,
65     C2, D1, D2, and S1 to S9 in the supplement). Overall, the outcomes did not noticeably change when the density correction was applied. Differences in the results scale with the density ratio applied (1.025).

This required following adaptions to the text:

- **Line 291:** Changed ORF value for BS3 from 0.43 to 0.42
- **Line 319:** Changed ORF value for SN12/1 from 0.45 to 0.44

70     **Line 181:** "The spatial distance of ca. 1 and 2.5 km" This isn't clear to me.

We clarified to which specific monitoring well(s) the respective distance referred to **(Line 215-218)**:

"The spatial distance between the meteorological station and the groundwater monitoring wells is approximately 1 km in case of SN12/1 and approximately 2.5 km in case of BS3 and NY-10. At this distance, the barometric pressure observations are assumed to be representative for the
75     groundwater monitoring locations as the barometric pressure typically varies at larger spatial scales (cf. Appendix A)."

**Line 418:** "conzeptualization" typo

Corrected, thanks!

**Response to report of referee #2, Rachel Housego**

I appreciate the author's efforts in revising their manuscript in response to reviewer comments and I look forward to seeing this in print at HESS. At this stage there is only one issue about the effect of wave set-up in the author's reply that needs to be resolved before this can be published. Due to the application of this paper specifically to coastal settings I think it is important that this is clarified in the final version of the manuscript. See full response below to the original discussion.

[Authors' note: following citation in referee report from previous round of revisions marked in italic and indented]

> *Neglecting wave set-up likely causes an issue in removing the oceanic effects on water levels, especially during surges. For more see the following papers and references therein. 565*
> *da Silva, P. G., Coco, G., Garnier, R., & Klein, A. H. (2020). On the prediction of runup, setup and swash on beaches. Earth-Science Reviews, 204, 103148.*
> *Stockdon, H. F., Holman, R. A., Howd, P. A., & Sallenger Jr, A. H. (2006). Empirical parameterization of setup, swash, and runup. Coastal engineering, 53(7), 573-588.*
>
> *We agree that waves may affect the oceanic response function. However, due to the high frequency of wave action, these effects would have very little to no effect on the data observed at the monitoring wells in this study. This is due to the low-pass filtering of the sediment which cancels the influence of high-frequency wave action over propagation distance. Wave setup can contribute significantly to groundwater table overheight also induced by tidal motion (Nielsen, 1990; 1999), but the oceanic response function focusses on the time-series dynamics rather than more persistent offsets such as is caused by wave-induced overheight. We included an explanation with a recommendation to include wave-setup data when analyzing groundwater-level time series close to the shoreline (Lines 237-242): "Besides ocean tides, waves can have a pronounced impact on near-shore groundwater-level dynamics (e.g., Nielsen, 1999; Housego et al., 2021). Due to the generally high-frequency of the wave dynamics at the shoreline (e.g., Stockdon et al., 2006; Hegge and Masselink, 1991) and the low-pass filter properties of the aquifer sediment (e.g., Rotzoll et al., 2008; Trefry and Bekele, 2004), waves can be assumed not to impact the groundwater-level dynamics at the monitoring wells in this study, which are several hundreds of meters from the shoreline (cf. Table 1). However, the influence of wave dynamics on groundwater levels may be relevant at beach sites or sites closer to the shoreline."*

I agree with the authors that the high frequency effects of wave action would not affect the inland groundwater levels. However, the net effect of wave set-up during storms is a long-term modification of the **mean water level at the shoreline due to conservation of momentum from wave breaking** which actually has a long effective wave period and would not be attenuated as described above and therefore could impact inland levels. **It is not a wave-by-wave process.** For example, if there were waves at 5 m offshore for a 2-day period during that entire two-day duration the water level at the shoreline would be elevated 1-1.25 m above the level predicted by using an offshore wave buoy to design the ocean time series. During calm conditions you can neglect this effect but during the storm responses this becomes important. The effect of set-up does attenuate inland and likely becomes less significant beyond 500 m inland where your sites are located. However, this is a methods paper specific to coastal settings so I think it is really important to present this accurately because this could be transferred to other coastal sites where wave setup would be important in terms of designing an accurate ORF, especially for time series where multiple surge events are being removed.

Agreed, the influence of wave setup should be given consideration in the conceptual and methodological sections of the manuscript. Therefore, we added and reformulated parts of Section 2.1 to give the wave influence more prominence in the conceptualization of the method **(Lines 52-62)**:

> "Sea-level variation is dominated by diurnal and semi-diurnal periodicities, along with aperiodic behavior resulting from storm events (Boon, 2011). Further, waves breaking at the shore impact groundwater-level dynamics (e.g., Nielsen 1999, Housego et al., 2021). Wave dynamics generally occur at high-frequencies at the shoreline (e.g., Stockdon et al., 2006; Hegge and Masselink, 1991) while the continuous wave breaking at the shore results in a more persistent, lower-frequency wave setup (Stockdon et al., 2006; Gomes da Silva, 2020). Wave setup is generally larger during storm events (Senechal et al., 2011) and thus adds to the magnitude of the storm-event related, aperiodic rises in sea level.

> The influence of fluctuating sea levels and waves diminishes with distance from the shoreline, with tidal and high-frequency wave variation attenuating more rapidly than variation from season, wave setup or extreme events, such as floods or droughts (Ferris, 1952; Li et al., 2004; Nielsen, 1990; Li et al. 1997; Carwright et al., 2006; Rotzoll and El-Kadi, 2008). Precipitation recharges groundwater by vertical percolation through the overlying unsaturated zone or by direct recharge from surface-water bodies that fill during storm events."

These paragraphs also replace Lines 237 to 242 in the submitted manuscript in large parts. This paragraph was reformulated accordingly **(Lines 274-279)**:

> "Wave setup was not considered as a separate process since the additional considerations required for an empirical formula to estimate wave setup from offshore measures (Gomes da Silva et al. 2020) were beyond the methodological objective of this technical note. The influence of wave setup on groundwater levels may however be present in the corrected time series when the wave setup present at calm conditions increases during storm events for example (i.e. wave setup is not constant over the studied time frame; cf. Section 2.1). Here, this could be the case during the storm event in January 2019 or the time frame of pronounced sea-level variations in March 2019 for example (Fig. 3)."

In **Line 257** we added: "and wave setup."

In Section 2.4. we added in **Lines 146-149:**

> "A wave response function and groundwater levels with wave setup removed can be obtained equivalently, e.g. to account for additional storm-event related wave setup at the shore. Alternatively, wave setup can be incorporated into the sea-level time series to obtain an ORF representing both processes. Note that wave setup is generally estimated from offshore wave measures by means of empirical formulas (c.f., Gomes da Silva et al., 2020)."

**Further changes and technical corrections**

**Lines 29 to 35:** Reformulated due to additional literature: "Convolution by means of transfer function noise modeling has been applied by Bakker and Schaars (2019) to model hydraulic heads of a coastal aquifer based on time series from sea level, recharge, and groundwater withdrawal. An estimation of a response function from sea-level data itself and removal of sea-level influences from dynamic

groundwater levels in coastal settings, like done with regression deconvolution, has not been performed (to the authors' knowledge). Especially in coastal settings periodic and aperiodic influences often obscure important groundwater processes, such as recharge, which is difficult to estimate or directly measure, and pumping."

**Line 66:** Replaced "ocean tide signal" with "sea-level signal".

**Lines 140 to 141:** Replaced "$\Delta x = \{\Delta SL, \Delta BP\}$" with "processes $p = \{SL, BP\}$ in Eq. (9).". The previous notation was from a previous notation of the processes that was changed before submission of the manuscript.

**Line 187:** Replaced "Norderney" with "Norderney's".

**Line 271:** Changed "temporally variability" to "temporal variability".

**Line 303:** Added "in the supplement" to figure references referencing to the supplement.

**Line 332:** Replaced "collected" with "available" for consistency with caption of Fig. 7.

**Line 441:** Added "in the supplement" to figure references referencing to the supplement.

**Lines 463 to 464:** Added "Further, we thank the two reviewers for their constructive comments which helped to significantly improve the paper."

**Figure 2:** Changed "meteorologic station" to "meteorological station" in caption.

**Figure 5:** Added "in the supplement" to figure references referencing to the supplement.

**Table 1:**

- Put "a" before DEM in footnotes (b) and (c)
- replaced "closest" with "nearest" in footnote (d)
- Corrected "top of screen [m asl]" for BS3 from -4.68 to -4.98
- Corrected "Distance to MHW [m]" for NY-10 from 978 to 979

**References:** Journal names were changed to Journal name abbreviations where not used beforehand.

**References**

Boon, J. D. (2011). Secrets of the Tide: Tide and Tidal Current Analysis and Applications, Storm Surges and Sea Level Trends. Cambridge: Woodhead Publishing. https://doi.org/10.1016/B978-1-904275-17-6.50011-2

Cartwright, N., Baldock, T. E., Nielsen, P., Jeng, D.-S., & Tao, L. (2006). Swash-aquifer interaction in the vicinity of the water table exit point on a sandy beach. Journal of Geophysical Research, 111(C9), C09035. https://doi.org/10.1029/2005JC003149

Ferris, J. G. (1952). Cyclic fluctuations of water level as a basis for determining aquifer transmissibility (USGS Unnumbered Series No. Note 1). Washington, D.C.: U.S. Geological Survey. https://doi.org/10.3133/70133368

Gomes da Silva, P., Coco, G., Garnier, R., & Klein, A. H. F. (2020). On the prediction of runup, setup and swash on beaches. Earth-Science Reviews, 204, 103148. https://doi.org/10.1016/j.earscirev.2020.103148

Greskowiak, J., & Massmann, G. (2021). The impact of morphodynamics and storm floods on pore water flow and transport in the subterranean estuary. Hydrological Processes, 35(3), e14050. https://doi.org/10.1002/hyp.14050

Grünenbaum, N., Günther, T., Greskowiak, J., Vienken, T., Müller-Petke, M., & Massmann, G. (2023). Salinity distribution in the subterranean estuary of a meso-tidal high-energy beach characterized by Electrical Resistivity Tomography and direct push technology. Journal of Hydrology, 617, 129074. https://doi.org/10.1016/j.jhydrol.2023.129074

Hegge, B. J., & Masselink, G. (1991). Groundwater-Table Responses to Wave Run-up: An Experimental Study From Western Australia. Journal of Coastal Research, 7(3). Retrieved from https://journals.flvc.org/jcr/article/view/78520

Housego, R., Raubenheimer, B., Elgar, S., Cross, S., Legner, C., & Ryan, D. (2021). Coastal flooding generated by ocean wave- and surge-driven groundwater fluctuations on a sandy barrier island. Journal of Hydrology, 603, 126920. https://doi.org/10.1016/j.jhydrol.2021.126920

Li, L., Barry, D. A., Parlange, J.-Y., & Pattiaratchi, C. B. (1997). Beach water table fluctuations due to wave run-up: Capillarity effects. Water Resources Research, 33(5), 935–945. https://doi.org/10.1029/96WR03946

Li, L., Cartwright, N., Nielsen, P., & Lockington, D. (2004). Response of Coastal Groundwater Table to Offshore Storms. China Ocean Engineering, 18(3), 423–431.

Nielsen, P. (1990). Tidal dynamics of the water table in beaches. Water Resources Research, 26(9), 2127–2134. https://doi.org/10.1029/WR026i009p02127

Nielsen, P. (1999). Groundwater Dynamics and Salinity in Coastal Barriers. Journal of Coastal Research, 15(3), 732–740.

Post, V., Kooi, H., & Simmons, C. (2007). Using Hydraulic Head Measurements in Variable-Density Ground Water Flow Analyses. Ground Water, 45(6), 664–671. https://doi.org/10.1111/j.1745-6584.2007.00339.x

Post, V. E. A. (2012). Electrical Conductivity as a Proxy for Groundwater Density in Coastal Aquifers. Ground Water, 50(5), 785–792. https://doi.org/10.1111/j.1745-6584.2011.00903.x

Rotzoll, K., & El-Kadi, A. I. (2008). Estimating hydraulic properties of coastal aquifers using wave setup. Journal of Hydrology, 353(1), 201–213. https://doi.org/10.1016/j.jhydrol.2008.02.005

Senechal, N., Coco, G., Bryan, K. R., & Holman, R. A. (2011). Wave runup during extreme storm conditions. Journal of Geophysical Research: Oceans, 116(C7). https://doi.org/10.1029/2010JC006819

Stockdon, H. F., Holman, R. A., Howd, P. A., & Sallenger, A. H. (2006). Empirical parameterization of setup, swash, and runup. Coastal Engineering, 53(7), 573–588. https://doi.org/10.1016/j.coastaleng.2005.12.005